# Oligomerisation of Ku from *Mycobacterium tuberculosis* promotes DNA synapsis

Sayma Zahid [1], Sonia Baconnais[2], Henrietta Smith[3], Saseela Atwal[1], Lucy Bates[4], Harriet Read[3], Ankita Chadda[5,9], Florian Morati [6], Tom Bedwell[7], Emil G. P. Stender [7], Joanne Walter[7], Steven W. Hardwick [8], Fredrik Westerlund [6], Eric Galburt [5], Eric Le Cam [2], Alice Pyne [3], Galina V. Mukamolova [4] & Amanda K. Chaplin [1] ✉

*Mycobacterium tuberculosis* (*Mtb*), the causative agent of tuberculosis (TB), is estimated to infect nearly one-quarter of the global population. A key factor in its resilience and persistence is its robust DNA repair capacity. Non-homologous end joining (NHEJ) is the primary pathway for repairing DNA double-strand breaks (DSBs) in many organisms, including *Mtb*, where it is mediated by the Ku protein and the multifunctional LigD enzyme. In this study, we demonstrate that Ku is essential for mycobacterial survival under DNA-damaging conditions. Using cryogenic electron microscopy (cryo-EM), we solved high-resolution structures of both the apo and DNA-bound forms of the Ku-*Mtb* homodimer. Our structural and biophysical analyses reveal that Ku forms an extended proteo-filament upon binding DNA. We identify critical residues involved in filament formation and DNA synapsis and show that their mutation severely impairs bacterial viability. Furthermore, we propose a model in which the C-terminus of Ku regulates DNA binding and loading and facilitates subsequent recruitment of LigD. These findings provide unique insights into bacterial DNA repair and guide future therapeutics.

In all kingdoms of life, DNA damaging agents can jeopardize the integrity of the cellular genome, resulting in a range of developmental defects. The intracellular human pathogen *Mycobacterium tuberculosis* (*Mtb*) causes the respiratory disease Tuberculosis (TB), a disease where an estimated one quarter of the global population has been infected, where in some cases the bacteria may persist in infected humans for decades[1]. A key reason for the pathogen's resilience within mammalian hosts is due to its ability to efficiently repair potentially cytotoxic damage to its DNA genome. Specifically inhibiting these DNA repair mechanisms therefore may be a potential therapeutic target. Non-

homologous end joining (NHEJ), is the major repair pathway for double strand breaks (DSBs) in DNA[2], which if left unrepaired can ultimately result in cell death. The NHEJ mechanism in bacteria is a simplified system compared to the human counterpart, as it is reliant on the action of only two proteins: Ku and Ligase D (LigD). Multiple studies have shown the ability of Ku to bind double-stranded DNA ends, interact with LigD and stimulate LigD repair activities[3]. In bacteria the LigD protein is responsible for the essential nuclease, polymerase and ligase activities. LigD contains three domains: polymerase, nuclease and ligase[4]. Structures of the ligase and polymerase domains of LigD

[1]Leicester Institute for Structural and Chemical Biology, Department of Molecular and Cell Biology, University of Leicester, Leicester, UK. [2]Genome Integrity and Cancers UMR 9019 CNRS, Université Paris-Saclay, Gustave Roussy, Villejuif Cedex, France. [3]School of Chemical, Materials and Biological Engineering, University of Sheffield, Sheffield, UK. [4]Leicester Tuberculosis Research Group, Department of Respiratory Sciences, University of Leicester, Leicester, UK. [5]Department of Biochemistry and Molecular Biophysics, Washington University School of Medicine, 660 South Euclid Avenue, Saint Louis, MO, USA. [6]Division of Chemical Biology, Department of Life Sciences, Chalmers University of Technology, Gothenburg, Sweden. [7]Fida Biosystems Aps, Generatorvej 6, 1st, Soeborg, Denmark. [8]Department of Biochemistry, University of Cambridge; Sanger Building, Tennis Court Road, Cambridge, UK. [9]Present address: Salk Institute for Biological Studies, La Jolla, CA, USA. ✉e-mail: ac853@leicester.ac.uk

have been solved by X-ray crystallography, in addition to the phosphoesterase domain from *Pseudomonas aeruginosa*, but the full structure of this protein has not been determined[5–7]. Additionally, no structural information is available for the Ku protein from any bacterial species[6,8]. Multiple methods have indicated a direct interaction between Ku and LigD but so far, no structural details of this complex have been obtained[9,10].

Understanding the fundamental process of DNA-repair in *Mtb* will help to determine how this mechanism is important for the bacteria to survive and cause infection, and how best to target this system for the design of future antimicrobials. Although an intensive drug regimen has been implemented for patients infected with *Mtb*, antimicrobial drug resistance has been a major problem and has become increasingly more prevalent[11]. Due to the advancement in the technology and methodology associated with cryo-electron microscopy (cryo-EM) in the past few years, the molecular understanding of NHEJ in humans has significantly advanced[12–16]. However, although NHEJ in bacteria is a simplified system utilising fewer protein components, the bacterial NHEJ mechanism and protein interplay are not well understood.

It has been shown that NHEJ in mycobacteria may be particularly important for bacterial survival whilst in stationary phase[17]. Previous studies have established that Ku and LigD are dispensable for mycobacterial growth in vitro and in infected animals[18] but may play a role in infection of macrophage[19]. Moreover, both proteins play a distinct role in survival of mycobacteria exposed to oxidative agents and desiccation[17], stresses likely encountered by mycobacteria during intracellular infection and transmission. However, further mycobacterial survival experiments are required to determine the specific importance of these NHEJ proteins for the bacteria.

In this study, we show with functional survival assays that Ku is essential for *Mtb*'s resilience under DNA-damaging conditions. To further investigate the NHEJ mechanism in *Mtb*, we present cryo-EM structures of the Ku protein, both in its apo form and in complex with DNA. We resolved the apo-Ku-*Mtb* structure to 4.04 Å resolution and the DNA-bound structure to 2.96 Å resolution. Using a combination of mass photometry, Flow Induced Dispersion Analysis (FIDA), cryo-EM, Atomic Force Microscopy (AFM), and positive-stain imaging, we demonstrate that Ku-*Mtb* forms higher-order filament structures in the presence of DNA. We also specifically identify residues important in Ku DNA synapsis and demonstrate their role in mycobacterial survival under DNA damaging conditions. Finally, we propose a model for NHEJ in *Mtb* and discuss how these insights could reveal key properties from human NHEJ, highlighting the potential for targeting bacterial NHEJ in the development of antimicrobials.

## Results

### Ku is important for mycobacterial survival under exposure to methyl methanesulfonate and dessication

Ku proteins are present in all mycobacteria with the exception of *Mycobacterium leprae* where *ku* is a pseudogene. Ku genes are non-essential for growth but have been proposed to be important for survival after exposure to ionising radiation and for stress response in the stationary phase[9,20–22]. To investigate this further we have made an unmarked *ku* deletion mutant(Δ*ku*) of *Mycobacterium smegmatis*, a fast-growing model mycobacterium. We observe no difference in growth or colony morphology of wild type and mutant strains under standard conditions in 7H9 medium. However, exposure of *M. smegmatis* strains to 0.1% (v/v) of methyl methanesulfonate (MMS), an inducer of double strand breaks, revealed a significant survival defect of Δ*ku* (Fig. 1A). Upon a 6-h exposure, colony-forming unit (CFU) counts of wild type decreased by 1.75 $\log_{10}$ compared with 3.5 $\log_{10}$ reduction of CFU counts of Δ*ku*. Importantly, reintroduction of *M. smegmatis ku* (msmeg_5580) restored the wild type phenotype (Fig. 1A).

*Mtb* is transmitted via aerosolised droplets and survival of desiccation is believed to be important for *Mtb* transmission. *M. smegmatis* also experiences desiccation stress in the environment. Figure 1B shows that desiccation of *M. smegmatis* indeed led to a dramatic loss of viable cells in all strains, nearly 2.6 $\log_{10}$ and 4.6 $\log_{10}$ reduction of CFU counts in wild type and Δ*ku*, respectively. The survival defect of Δ*ku* was complemented by reintroduction of *M. smegmatis ku* (Fig. 1B). These findings confirm that Ku is critical for mycobacterial survival under conditions that promote double strand DNA breaks.

### The structural architecture of apo-Ku-Mtb

To structurally analyse Ku-*Mtb* the purified apo protein (Supplementary Fig. 1) was applied to cryo-EM grids in conditions similar to those used successfully for human Ku70/80 samples[14]. Given the relatively small molecular weight of Ku (~60 kDa dimer) we were unsure whether cryo-EM would be suitable to resolve a high-resolution structure of this protein. However, data was collected from a single grid and following data processing a final map of homodimeric Ku was obtained at 4.06 Å resolution with C2 symmetry imposed (Fig. 1C, D, Supplementary Figs. 2, 3, Supplementary Table 1).

Two copies of the structural model of Ku-*Mtb* monomer as predicted by AlphaFold 2 could be docked into the experimental cryo-EM map, requiring minimal further refinement (Fig. 1D). Protomer A and B of Ku-*Mtb* can be seen to intertwine in order to form a central positively charged DNA binding pocket, similar to the DNA binding channel formed by human Ku70/80. When compared to human Ku70/80, the most significant difference is the absence of the vWA domains present in Ku70 and Ku80 (Fig. 1E, F). Significantly, the AlphaFold 2 model predicted the C-terminal α-helix, which is flexibly linked to the core dimer could bind to the outer surface of Ku-*Mtb*, whilst AlphaFold 3 modelled this helix not engaged to the core of Ku-*Mtb* (Fig. 1D, F and Supplementary Fig. 4). Our structural data confirms the location of the AF2 predicted interaction site.

### Ku-Mtb in the presence of DNA forms oligomers

To determine the mechanistic details of Ku-*Mtb* biding to DNA, we added a DNA substrate containing a 32 bp double-strand and 20 nt single-strand (SS), previously utilised for UvrD1 and Ku studies[23]. Using this DNA previously, multiple Ku molecules binding was observed with high cooperativity, with the best model fitting to 2.5/3 Ku/DNA[23]. Electromobility shift assays (EMSAs) showed a band of shifted DNA upon increasing concentrations of Ku. At a ratio of 1:4 (DNA:Ku) a stable shifted band was observed and thus cryo-EM grids were prepared using the protein:DNA complex at this ratio (Fig. 2A). Strikingly, we observed clear extended linear and branched filaments within the cryo-EM micrographs (Fig. 2B). Following cryo-EM data collection and processing we could clearly visualise these filaments within the 2D class averages (Fig. 2C).

Following data processing we subsequently obtained a map to 2.96 Å resolution, into which we could model a filament of Ku-*Mtb* bound to DNA (Fig. 2D, Supplementary Figs. 5, 6, Supplementary Table 1). This filament is continuous with a minimal repeating unit of two Ku-*Mtb* homodimers bound to one DNA duplex, in agreement with our EMSA experiments indicating a stable complex of four Ku protomers with 1 DNA duplex (Fig. 2D). Within our cryo-EM map, we have modelled 1.5 repeating units (3 Ku-*Mtb* homodimers and two DNA molecules). The central DNA binding cradle of Ku-*Mtb* threads on to DNA, in a similar manner to human Ku70/80. Neighbouring repeating units position the two DNA ends at a distance of ~40 Å apart. No density for the single-stranded DNA overhang can be identified within our cryo-EM map. Given the high quality of the experimental map, we were able to visualise density corresponding to a potential metal ion (likely to be Mg due to $MgCl_2$ in the buffer) coordinated by residues His27, Asp26, Asp28 from one protomer and Trp169, Glu172 from another protomer, which is at the interface between two Ku protomers

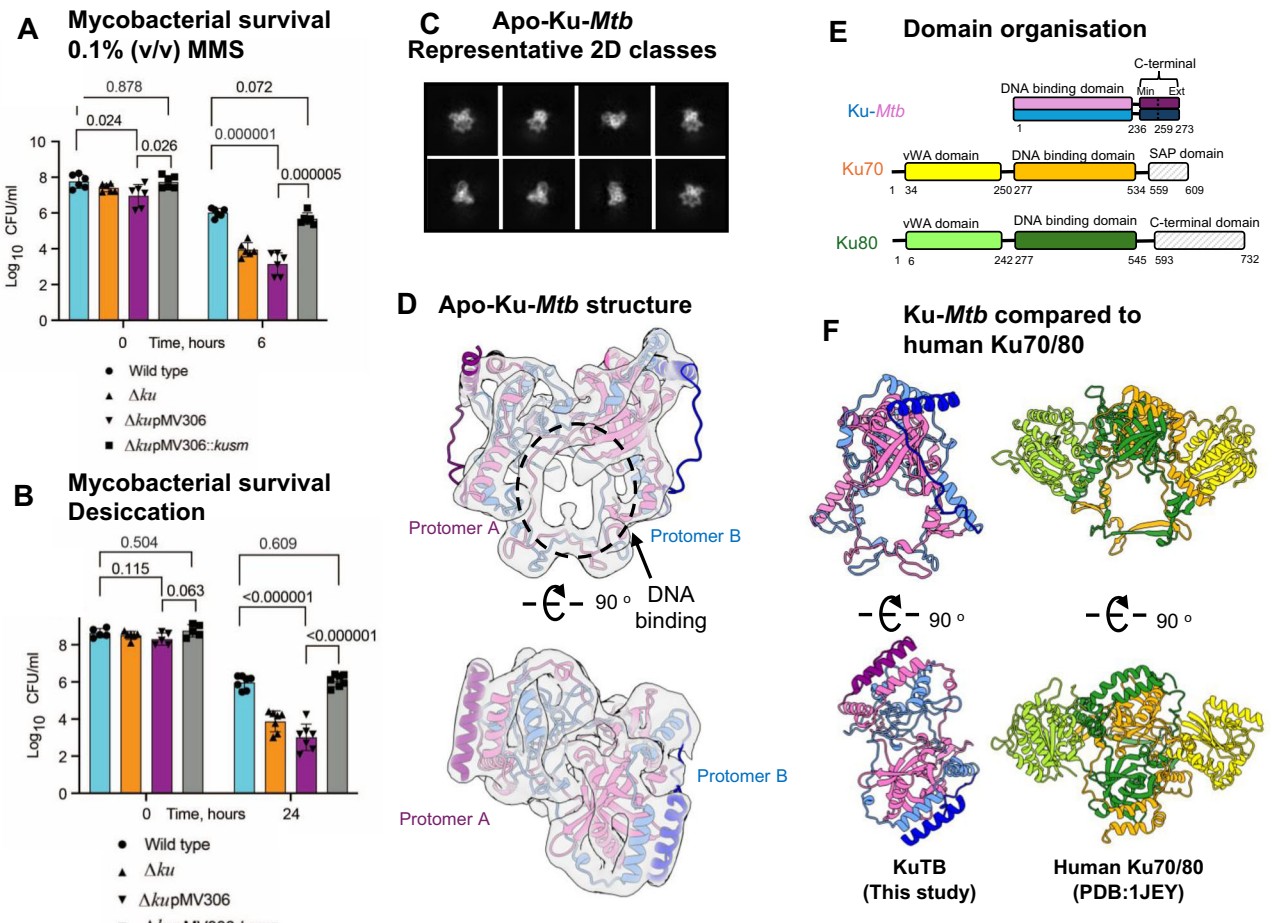

**Fig. 1 | The importance of Ku for mycobacterial survival under DNA double-strand break conditions and cryo-EM of Apo-Ku-*Mtb*.** *M. smegmatis* strains from stationary phase were exposed to 0.1% (v/v) MMS (**A**) or subjected to desiccation for 24 h. **B** CFU counts were determined before and after treatments on 7H10 agar. Δ*ku* – *ku* deletion mutant, Δ*ku*pMV306 - *ku* deletion mutant with empty plasmid; Δ*ku*pMV306::*kusm* – complemented *ku* deletion mutant containing pMV::*msmeg5580*. Results of two independent experiments, done with three biological replicates. ns – non-significant, **** *p* < 0.0001, one way ANOVA or unpaired *t* test. **C** Representative 2D class averages of apo-Ku-*Mtb*. **D** Cryo-EM map and model for apo-Ku-*Mtb*. The map is shown as a transparent grey to 4.04 Å resolution. Protomer A is shown in pink and protomer B in blue, the C-terminal extensions are shown in darker colours. **E**) Domain organisation schematic for Ku-*Mtb* and human Ku70/80. Ku-*Mtb* is shown in pink and blue. Ku70 vWA is shown in yellow and the DNA binding domain in orange. Ku80 vWA is coloured in lime green and the DNA binding domain in dark green. **F**) A comparison of Ku-*Mtb* (This study) and human Ku70/80 (PDB:1JEY). The models are coloured according to **E**.

but not within binding distance to the DNA substrate, therefore it is likely that this ion has a role in structural stability of the Ku homodimer. Another potential coordination site between Ku protomers can be seen at the bridge of Ku, coordinated by residues His35 from one protomer and Cys48 and Cys51 from another protomer, again likely to stabilise the homodimer. We have generated a model of an extended Ku-*Mtb* filament (Fig. 2E) by superposition of multiple copies of our cryo-EM model. A full helical turn of the filament is created by six Ku homodimers. We believe that the illustrated twist is more due to the short DNA lengths rather than Ku, and if the DNA lengths were longer, we would not observe such an effect. A direct overlay of apo-Ku-*Mtb* with DNA-Ku-*Mtb*, illustrates that there are minor changes in the overall structure upon. DNA binding (Supplementary Fig. 7). There appears to be a slight compaction of the structure upon DNA binding as previously proposed[24].

**Oligomerisation of Ku-Mtb with DNA**

To assess the oligomeric states of Ku-*Mtb* in solution we utilised both mass photometry, FIDA and atomic force microscopy (AFM) experiments. Mass photometry indicated the presence of a small population of higher order assemblies following addition of DNA and increasing

ratios of Ku to DNA (Fig. 3A and Supplementary Fig. 8). The relative low abundance of filaments in this experiment may be due to the low concentration of protein used compared to the cryo-EM grids (20–60 nM vs 66 μM).

We additionally utilised the FIDA 1 system to explore *Ku-Mtb* binding to DNA. In these experiments the DNA was fluorescently labelled (FAM), therefore we utilised the 480 nm detector, to detect the change in the hydrodynamic radius (Rh) as Ku concentrations were increased. The Rh of the DNA alone was ~3.5 nm, corresponding to that predicted (3.1 nm). Upon increasing concentrations of Ku-*Mtb*, the Rh could be seen to first decrease and then increase very dramatically (Supplementary Fig. 8). These two events were fitted to independent binding curves. Fitting produces an initial binding with an apparent Kd in the nM range (green line–Supplementary Fig. 8) assuming a 1:1 binding event. This decrease in Rh value could be due to the structural compaction upon DNA binding as shown in Supplementary Fig. 7. For the second binding event again assuming a 1:1 binding event gave an apparent Kd value in the μM range (purple line – Supplementary Fig. 8). We interpret this as an initial binding of Ku-*Mtb* to DNA being high affinity, followed by lower affinity oligomerisation of minimal Ku:DNA complexes. This continuous increase in Rh is indicative of the

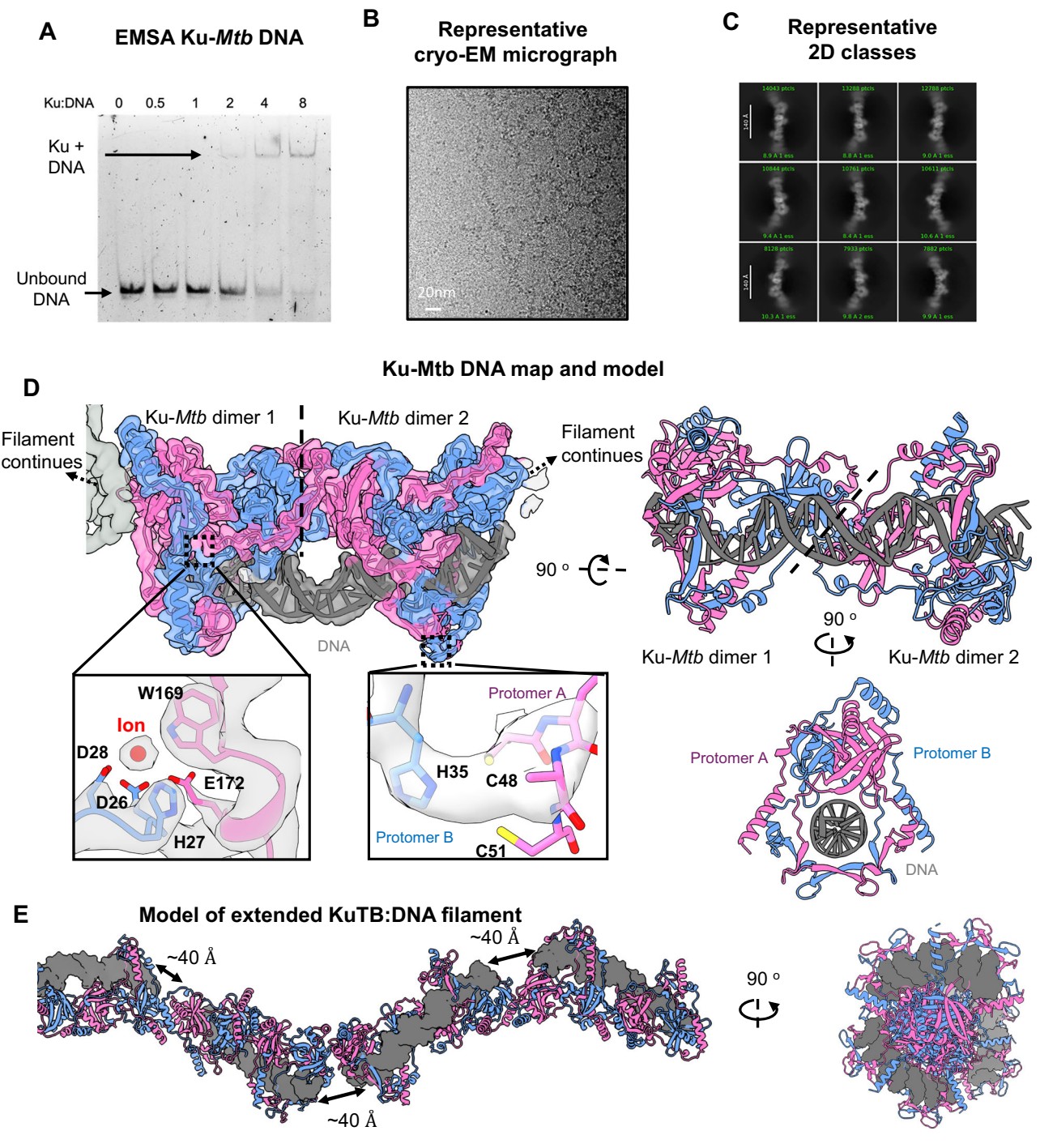

**Fig. 2 | Ku-*Mtb* binding to DNA. A** EMSA gel of DNA with increasing ratio of Ku protein. **B** Representative cryo-EM micrograph showing filaments of Ku-*Mtb*. **C** Representative 2D class averages. **D** Cryo-EM map and model of Ku-*Mtb* with DNA showing the repeating unit of two Ku homodimers bound to DNA. Protomer A in pink, protomer B in blue and DNA in grey and map at 2.96 Å resolution coloured according to chain colour. Insets show interfaces between protomers with one binding to metal ion. **E** Two views of an extended model of the Ku-*Mtb* filament, showing ~40 Å distance between DNA ends.

protein filament extending. We explored potential oligomerisation further and measured more data points with higher concentrations of Ku-*Mtb* (Fig. 3B). Again, upon increasing Ku concentrations, Rh increased. Interestingly, we observed specific plateau points in Rh as the Ku concentration was increased. The plateaus in the Rh value correspond to known predicted sizes for Ku-*Mtb* with DNA. The first plateau could be seen at ~5.5 nm (predicted ~6.1 nm), corresponding to the dimensions we observed in our cryo-EM data for complexes containing 3 Ku molecules and 2 DNA molecules. A plateau at ~6.3 nm (predicted ~7 nm) corresponds approximately to 4 Ku molecules and 2 DNA molecules. Upon increasing Ku concentration further, the Rh value continues to increase, indicative of further Ku oligomerisation (Fig. 3B). We therefore did not fit any binding curves to this data due to the dynamic and not 1:1 binding observed. Both mass photometry and FIDA techniques highlight the oligomerisation of Ku upon DNA binding.

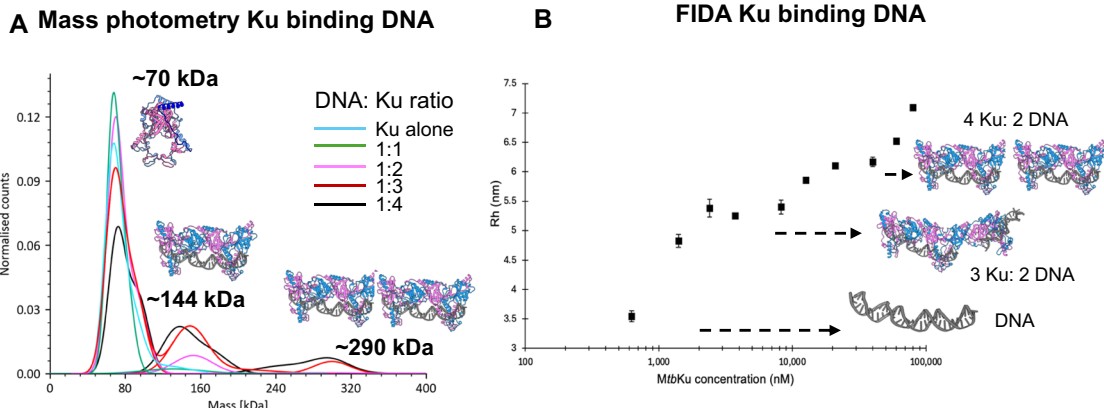

**Fig. 3 | Ku-*Mtb* binding DNA. A** Mass photometry of Ku binding to DNA, with increasing ratios of Ku:DNA and increasing molecular weight peaks observed with models above each shown. **B** FIDA experiment, showing the hydrodynamic radius (Rh) (nm) as Ku protein concentration is increased with DNA. No fits are added due to it not being a 1:1 binding event. Plateaus in the Rh value are shown how they correspond to the different cryo-EM structures observed.

Furthermore, we assessed the ability of Ku to form oligomers in the presence of DNA using AFM. Initially, AFM images of the blunt-ended and FAM-overhang DNA were analysed (Supplementary Fig. 9). These images confirm the presence of the single-stranded overhang in the FAM-labelled DNA substrate. We used the FAM-overhang DNA for all AFM experiments, observing the DNA molecules as single isolated species, of volume $63 \pm 8.7$ nm³, (mean ± SEM), (Supplementary Fig. 10Ai). The Ku-*Mtb* protein appeared as individual proteins with small numbers of larger globular aggregates, typically less than a few thousand nm³ ($231$ nm³ $\pm 36.6$; mean ± SEM) (Supplementary Fig. 10Aii).

To determine whether Ku oligomerisation occurred in the presence of DNA, the two were co-incubated at a ratio of 1:4 (DNA:Ku), as in our cryo-EM studies. Upon co-incubation with DNA, larger fibrillar and heterogeneous assemblies were visualised (Supplementary Fig. 10Aii and D) of volume 2466 nm³ $\pm 358.8$ (mean ± SEM), with many of the large structures formed exceeding 10,000 nm³ (Supplementary Fig. 10B). However, this value is likely underestimated due to masking that includes unbound Ku proteins, which lowers the overall mean. When tracing along these structures, we observe that they are up to 8 nm in height, around three times the height of a single Ku protein ($2.4 \pm 0.03$ nm; mean ± SEM) (Supplementary Fig. 10C). These assemblies are significantly larger in lateral size and volume than DNA or Ku alone (Supplementary Fig. 11) and indicate that only in the presence of DNA, is Ku able to form large assemblies. Interestingly, although AFM data may suggest a presence of branched filaments, we only observe linear assemblies using cryo-EM (Fig. 2).

To eliminate the potential influence of divalent cations in driving the observed Ku oligomerisation following DNA binding, AFM was also performed using an alternative, APS immobilisation method, which enables using buffers with no divalent ions (Supplementary Fig. 11). Large, fibrillar DNA-Ku complexes were still observed, with volumes ranging up to 125,000 nm³ in good agreement with data taken under the cryo-EM conditions, confirming that the observed oligomerisation is not due to the presence of divalent ions.

Notably, cryo-EM and AFM data may differ slightly, as these are distinct imaging modalities which can influence the observed structures. The AFM data shows large complexes that are filamentous, and in some areas appear more branched and clustered. The structures in cryo-EM appear longer, linear filaments, and this variation may be due to differences in sample preparation. In AFM, DNA and proteins are absorbed onto mica, which can promote altered or clustered conformations, especially in the case of flexible or dynamic assemblies like DNA-Ku. Whilst AFM provides topological data, it is limited in spatial resolution and makes it challenging to resolve the finer features of

filament assembly. In contrast, cryo-EM requires rapid vitrification and imaging under vacuum, which could favour more ordered conformations, and obscure dynamic or heterogeneous assemblies. Each technique provides complementary insight that, when. Integrated, allows for a better understanding of DNA-Ku oligomerisation.

### Ku protein interaction interfaces and disruption of the oligomer

Interestingly, within the DNA filament cryo-EM map, no density was observed for the C-terminal α-helix, as identified within the apo structure. Instead, it appears that the C-terminal helix has been displaced by either the action of DNA binding or the binding of Ku-*Mtb* at the filament interface. In fact, the positioning of this C-terminal α-helix within the apo structure is not possible when DNA is bound, as the helix interaction surface is occluded due to the filament formation (Fig. 4A, B).

We observe direct Ku-Ku protein interfaces within the filament, which is formed by two α-helices and two intertwined loops forming a largely hydrophobic interface (Fig. 4B). Although the loops containing Leu13 and Val14 are consistent throughout the filament, the helices containing residues Val198, Met202 and Met194 are displaced relative to each other at the junction between minimal repeating units (differing whether the interface is a gap in the DNA or engaging DNA) (Fig. 4B).

To identify residues critical for Ku oligomerisation and DNA synapsis, we performed site-directed mutagenesis targeting the Ku-Ku protein interface. Mutations introduced at the helical interface (residues Ser201 and Gln197) did not disrupt oligomer formation. We determined this by over-expression and purification of the double mutant S201A/Q197A. Following purification we froze cryo-EM grids of the protein, screened and collected a small data collection. The cryo-EM micrographs still showed the presence of Ku filaments, also identified in the 2D class averages, showing this interface is not important in oligomerisation (Supplementary Fig. 13).

However, introducing a double mutation (L13A/V14A) within the hydrophobic loop interaction interface abolished Ku oligomerisation on DNA (Fig. 4C−E). Following purification of the L13A/V14A mutant, we prepared cryo-EM grids using the same DNA and conditions as for the wild-type (WT) protein shown in Fig. 2. Initial screening of cryo-EM micrographs revealed a complete absence of filament formation. Subsequent short data collection and 2D classification further confirmed the lack of extended filaments. Despite this, we obtained a cryo-EM reconstruction of two Ku molecules bound to DNA at 3.67 Å resolution (Fig. 4E). This structure demonstrates that although the mutant Ku can still bind DNA, the Ku-Ku interaction required for filament formation across a DNA break is destabilised.

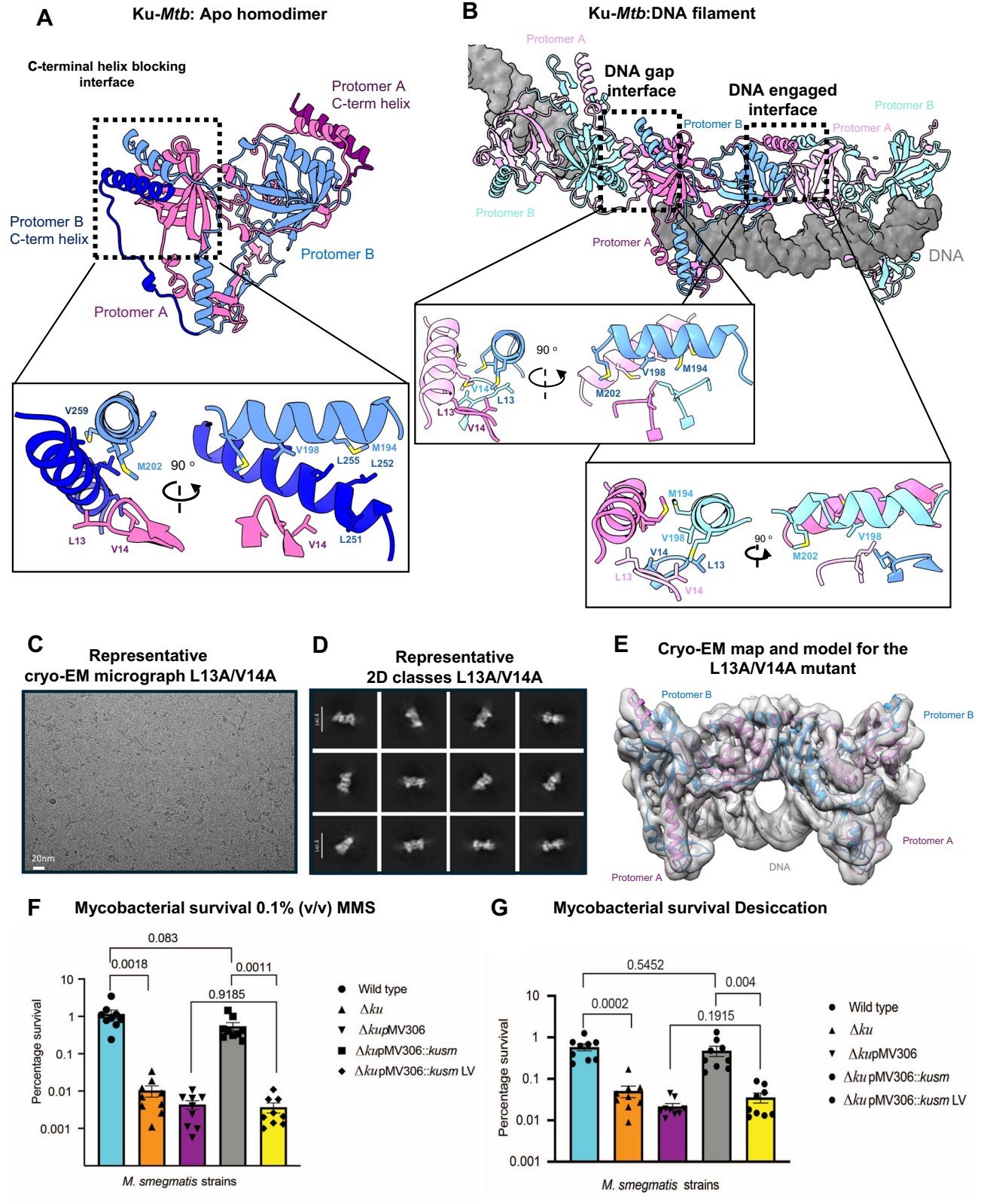

**A** Ku-*Mtb*: Apo homodimer

**B** Ku-*Mtb*:DNA filament

**C** Representative cryo-EM micrograph L13A/V14A

**D** Representative 2D classes L13A/V14A

**E** Cryo-EM map and model for the L13A/V14A mutant

**F** Mycobacterial survival 0.1% (v/v) MMS

**G** Mycobacterial survival Desiccation

Following identification of the specific residues involved in oligomerisation we carried out further in vivo experiments. The double mutation (L23A/V24A in *M. smegmatis* Ku protein equivalent to L13A/V14A *M. tuberculosis*) was introduced in the pMV306::*kusm* construct. *M. smegmatis* strains were either treated with 0.1% (v/v) MMS (Fig. 4F) or desiccation for 24 h (Fig. 4G). As previously observed *Δku* mutant with or without the empty pMV306 plasmid had significantly decreased survival percentages compared to WT in both conditions (Fig. 4F, G, supplementary source data file). While reintroduction of WT *M. smegmatis ku* gene fully complemented these survival defects, the L23A/L24A mutant (*Δku* pMV306::*kusm*LV)* showed decreased survival with survival

**Fig. 4 | Disruption of Ku oligomerisation. A** Apo-Ku-*Mtb* model with protomer A coloured pink, with the C-terminal linker and a-helix coloured purple, protomer B is coloured in blue with the C-terminal linker and a-helix coloured dark blue. Inset, shows the interface between the C-terminal a-helix and Ku which blocks filament formation in the apo structure. Residues are shown as sticks and labelled. **B** Ku-*Mtb* DNA filament model showing interfaces between Ku-Ku molecules. The central Ku homodimer is coloured pink for protomer A and blue for protomer B. The two Ku molecules either side are coloured light pink for protomer A and light green for protomer B. The DNA is coloured grey. Inset left, shows a close-up view of the Ku interface at the gap between the DNA ends, with residues shown as sticks and labelled. Inset right, shows a close-up view of the Ku interface between the DNA engaged interface with residues shown as sticks and labelled. **C** Representative cryo-EM micrograph for the L13A/V14A Ku-*Mtb* double mutant. **D** Representative 2D class averages for the L13A/V14A Ku-*Mtb* double mutant. **E** Cryo-EM map and model for the L13A/V14A Ku-*Mtb* double mutant to 3.67 Å resolution. Protomer A is coloured pink and protomer B blue, with the DNA shown in grey. **F** Survival percentages of *M. smegmatis* strains exposed to 0.1% (v/v) MMS for 6 h or **G**) desiccated for 24 h. Δ*ku* – *ku* deletion mutant, Δ*ku*pMV306 - *ku* deletion mutant with empty plasmid; Δ*ku*pMV306::*kusm* – complemented *ku* deletion mutant containing pMV::*msmegS580*, Δ*ku*pMV306::*kusm LV* – *ku* deletion mutant containing pMV::*msmegS580* with L23A/V24A mutations. Percentage survival was calculated by dividing CFU ml$^{-1}$ after treatment by CFU ml$^{-1}$ before treatment, multiplied by 100. Results of three independent experiments, performed with three biological replicates, error bars show SEM. ns – non-significant, *** $p < 0.001$, ** $p, 0.01$, unpaired t-test, unpaired *t* test; ns – non-significant ($p > 0.05$, one way ANOVA or unpaired t test).

percentages comparable to those of Δ*ku* and Δ*ku* pMV306 ($p > 0.05$, one way ANOVA). These findings confirm that Ku filament formation is important for DNA repair and mycobacterial survival under DNA damaging conditions.

### Ku-Mtb filaments formation is not dependent on the DNA end configuration

Although filaments of DNA-Ku-*Mtb* could be visualised, no density for the DNA SS-overhang could be identified within the cryo-EM map. To determine whether the SS-overhang is important for filament formation we first carried out EMSA studies to determine Ku binding with a blunt-ended DNA. Similar to with the previous DNA utilised, a shifted band could be observed upon increasing Ku concentration (Supplementary Fig. 14). We, therefore, froze cryo-EM grids under the same conditions as used previously but instead with the addition of blunt-ended DNA. Similar to the previous cryo-EM micrographs we could visualise filaments of Ku with blunt DNA. Following a small data collection of 1000 movies, we were able to reconstruct a cryo-EM map to 4.55 Å resolution (Supplementary Fig. 15). When overlaid with the previous map (shown in Fig. 2), the maps and subsequent models appear almost identical. This therefore shows that oligomerisation of Ku-*Mtb* is not dependent on the DNA end configuration.

Furthermore, we also froze cryo-EM grids under the same conditions as previously used but with addition of hairpin DNA (Supplementary Fig. 13). These micrographs displayed Ku-*Mtb* filaments as previously identified, again confirming that oligomerisation is not dependent of DNA end configuration.

### Positive staining EM reveals Ku-Mtb can circularize DNA

Positive staining EM was subsequently performed on Ku-*Mtb* with DNA to confirm the presence oligomers of Ku-*Mtb*, and additionally to assess the ability of Ku-*Mtb* to recircularise larger DNA substrates (as a proxy for synapsing DNA ends during NHEJ). We show that Ku-*Mtb* is able to load onto the DNA extremities and promote end-to-end joining between two DNA ends (Fig. 5). This was observed with blunt and 3' overhang DNA ends. We also found that Ku-*Mtb* loading onto DNA is cooperative and is magnesium dependent as previously suggested[23]. We varied the concentration of Ku-*Mtb* to observe protein loading and found that at 30/50 nM, very few DNA ends (401 or 1440 bp) were recognised by Ku. When the Ku protein concentration is increased to 100 and 200 nM, we observed that the percentage of Ku-DNA complexes compared to free DNA is greater than 80 %. In both cases, we observed a similar loading of Ku onto the DNA, with ~50 bp covered in Ku protein at each DNA end (Fig. 5). Such DNA ends recognition by Ku-*Mtb* with threading events to form filaments was previously observed by same EM approaches for Ku *Bacillus subtilis*[25,26].

What was particularly striking is the exceptional ability of Ku-*Mtb* to circularize linear DNA fragments (1440 or 401 bp) (Fig. 5, Supplementary Fig. 15). At 200 nM Ku-*Mtb*, more than 80 % of complexes lead to circularization and around 65 % at 100 nM. In both cases, 5% of complexes have both ends linked when 10-15 % and 25-30 % have only

one end occupied at 200 and 100 nM, respectively (Supplementary Table 2). Such observations clearly indicate that when both ends are recognized by Ku-*Mtb*, circularization occurs. This circularization can occur in two ways: end to end joining with a continuity of Ku-*Mtb* filaments along DNA, presumably in an arrangement similar to our cryo-EM structure. Additionally, we observe bridging events between two filaments side by side, which can be distinguished in two configurations: either the two filaments are in the same orientation leading to a pinching of the circle called B1 or in opposite orientation leading to a regular circle form, called B2 (Fig. 5, Supplementary Table 2). The average DNA lengths measured on the100 blunt molecules exhibiting end-to-end joining events (Fig. 5C) were increased by an average of 3.8 nm ± 0.8 compared to linear complexes with both ends occupied by Ku. This is in agreement with Cryo-EM results showing a distance of 40 Å between the two ends bridged by the filament.

Furthermore, we have characterised Ku-*Mtb* DNA binding properties with various configurations of DNA fragments, both in length and using a 3'overhang. We have subsequently analysed percentages of complexes which are either linear or circularised, the covered DNA sizes and discriminated the nature of circularization (Supplementary Table 2). Interestingly, similar affinities of Ku-*Mtb* for DNA were observed whether the DNA ends were blunt or contained an overhang (10 or 40 nt). For all these types of DNA fragments, the level of circularization is similar, i.e. above 80% of Ku-DNA (Supplementary Table 2). Additionally, for all DNA configurations (400 or 1440 bp, blunt or 3'overhang), we have also visualized linear or circular dimers (Supplementary Fig. 16).

Therefore, positive stain EM again confirms the ability of Ku-*Mtb* to oligomerize on DNA but also to circularize. We show that Ku-*Mtb* is able to keep DNA ends close via a synaptic complex, which no doubt promotes for LigD to interact and ligate the broken DNA ends. In contrast, Ku70/80 in humans is unable to hold broken DNA ends in close proximity and requires the binding of DNA-PKcs to form DNA-PK dimers[12,13]. Ku forms either end-to-end junctions or bridging of the DNA forming these synaptic assemblies. These two modes could be in equilibrium to promote ligase access.

## Discussion

We have determined the cryo-EM structures of Ku-*Mtb*, both in the presence and absence of DNA, revealing its capacity to form oligomers when bound to DNA. This DNA-induced oligomerization of Ku-*Mtb* is a unique observation, as human Ku70/80 cannot form such protein-protein interacting filaments on DNA. Our findings, supported by positive-stain EM, demonstrate that Ku-*Mtb* coats DNA ends and facilitates the synapsis of broken DNA ends. This mechanism enables the DNA ends to be held in close proximity, promoting subsequent DNA ligation by LigD.

### Proposed role for the conserved C-terminal helix in NHEJ

Data presented here also highlights the importance of the C-terminal region of Ku-*Mtb*. Ku proteins from mycobacteria show high sequence

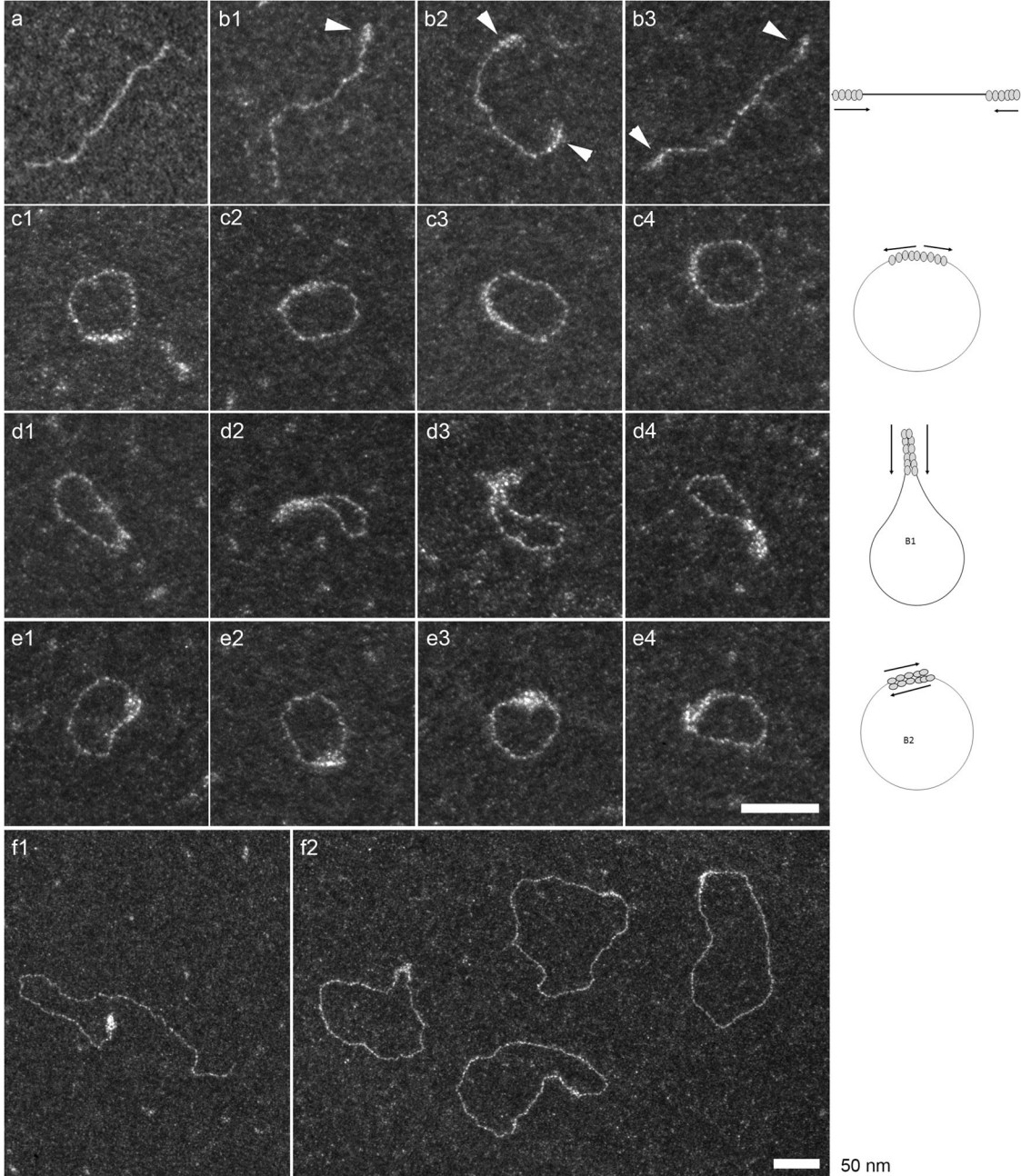

**Fig. 5 | Positive stain of Ku-*Mtb*.** Electron micrographs of DNA-Ku*Mtb* complexes obtained at 200 nM Ku with blunt 401 bp (**a-e**) and 1440 bp (**f**) fragments. The samples were analyzed in positive staining and darkfield imaging mode. (**a**) 440 bp control DNA fragments (**b**) Ku filaments at one end (b1) or at two ends (b2-3); The arrows indicate Ku filaments. (c 1–4) circularization of DNA fragment mediated by end-to-end joining. (d1-4) circularization of DNA fragment mediated by bridging B1 where the two filaments are in the same orientation. (e 1–4) circularization of DNA fragment mediated by bridging B2 where the two filaments are in opposite orientation. (f1) 1440 pb DNA fragment linked by Ku at one end and (f2) circularization of DNA fragments mediated by Ku. The two scale bars correspond to 50 nm.

similarity, especially the conserved core DNA binding region (Supplementary Fig. 17)[27]. Most bacterial Ku homologues have a C-terminal tail that varies in length between species, and even between mycobacterial species (Supplementary Fig. 17). Ku from *Mtb* has a relatively short C-terminus, especially when compared to *M. marinum* for example which has a longer C-terminal extension (Supplementary Fig. 17). It was shown by McGovern et al., that the inner minimal C-terminal region is responsible for recruitment of LigD in *Bacillus subtilis*[25,26]. They also showed that the C-terminal extension controls Ku's threading, slowing it down, which leads to an increase of Ku concentration near the DNA ends[25]. This was also investigated using Ku-*Mtb* by Sowa et al., where it was demonstrated that the Ku

C-terminus is a multi-purpose arm for binding DNA and stimulating ligation[28]. Here, through our cryo-EM and biophysical studies we propose a model for DSB repair through movement of the C-terminal helix of Ku (Fig. 6). In the absence of a bound DNA substrate, the C-terminal helix of Ku docks onto the surface of the protein, occluding the proteofilament interface. Once a DSB site is recognised, Ku binds onto the DNA ends and the C-terminal α-helix is released from the position identified in the apo structure. Given the proposed role of the C-terminal extension of Ku in binding of LigD to promote DNA ligation, we hypothesize that in the DNA engaged state the release of the C-terminal α-helix from the surface of Ku during filament formation will also promote binding and recruitment of LigD, stimulating repair

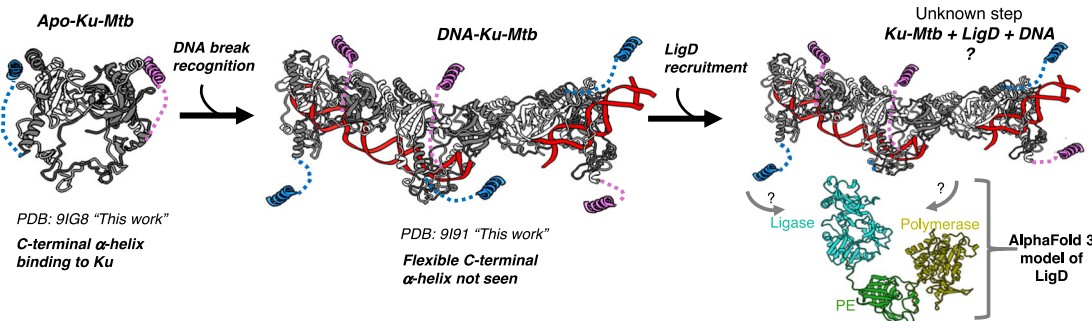

**Fig. 6 | Model of the NHEJ mechanism in *Mtb*.** Apo-Ku-*Mtb* is first shown in grey with the C-terminal helices coloured in blue and pink. Ku-*Mtb* then binds to DNA loading and forming an oligomer, with the C-terminal helices then released to allow filament formation. LigD is then finally recruited and DNA ligated in an unknown mechanism. An AlphaFold 3 model of LigD is shown and coloured according to three domains, ligase, PE and polymerase.

of the DSB (Fig. 6). This may also be a way in which to regulate Ku and LigD only interacting when DNA is present.

## Comparison of NHEJ in bacteria and humans

Invertebrates including, bacteria and yeast do not contain DNA-PKcs-like enzymes meaning an enormous number of organisms (more than the number of vertebrate organisms) carry out NHEJ without DNA-PKcs. Understanding how these species are able to perform this repair process is paramount to understanding NHEJ in invertebrates and may provide clues as to how and why this system evolved to the more complex arrangement utilised by humans.

From our structural and biophysical studies, we provide evidence for how the Ku homodimer is able to bind and join broken DSBs in isolation, an action which requires the DNA-PK holoenzyme in humans[12,13]. The observed filaments could therefore be a consequence of the strong Ku-Ku interactions in the absence of DNA-PKcs. The filaments resulting from the protein-protein interactions likely help maintain a higher local concentration of Ku near the DNA break ends. This makes them more stable, both to protect DNA ends and increase the recruitment of LigD. When comparing the structures of Ku70/80 from humans with the Ku homodimer in bacteria (Fig. 1 and Supplementary Fig. 18), the most notable differences is the absence of the vWA domains in bacteria and the homodimeric assembly. These vWA domain found in Ku70/80 in humans are critical for recruitment of canonical and regulatory proteins to the DSB during NHEJ. Specifically, it has been shown that the core protein XLF binds to the Ku80 vWA domain[13,29,30], whereas PAXX binds to the vWA domain of Ku70[14,31–34]. Additional NHEJ regulatory proteins including WRN, APLF and CYREN have also been shown to interact with the vWA domain of Ku80[29,35]. Furthermore, NHEJ in humans involve individual nucleases and polymerase enzymes, including the nuclease Artemis and Polymerases λ and μ which may be necessary depending on the type of DNA break. The absence of vWA domains in bacterial Ku allows for an alternative mechanism to recruit accessory proteins, specifically the C-terminal α-helix, which is released upon DNA binding and filament formation to allow for LigD recruitment. When comparing the structure of Ku70/80 from humans to DNA-Ku-*Mtb*, the position of the vWA domains do not occlude the filament binding interface (Supplementary Fig. 18). However, when comparing Ku70/80 with the Apo-Ku-*Mtb* structure, the Ku70/80 heterodimer has a helix engaged in a similar position to the C-terminal α-helix we observe in the Ku-*Mtb* structure, therefore would prevent filament formation (Supplementary Fig. 18). From our structural data it is clear that in Ku-*Mtb* this C-terminal helix must first dis-engage to allow filament formation. In all structures of human Ku70/80 to date a movement in this equivalent helix has not been observed, and even though Ku70/80 can load onto DNA, it seems unlikely that human Ku70/80 will be able to form filaments in a similar manner.

Furthermore, the positive stain EM data shows that Ku-*Mtb* is able to recircularize DNA, something that human Ku70/80 is unable to do (Fig. 4). It seems that Ku-*Mtb* is therefore able to synapse the broken DNA ends, ready for LigD to carry out ligation. This differs in humans as NHEJ requires DNA-PKcs to be present within the DNA-PK holoenzyme complex, forming DNA-PK dimers to enable synapsis of the DNA ends[12,13]. Interestingly, we have observed higher order oligomers of DNA-PK, including trimeric, tetramers and higher[16]. It is intriguing that we observed higher order NHEJ assemblies in both humans and bacteria. One may be concerned of the high protein concentrations utilised in cryo-EM experiments, however firstly Ku is a highly abundant protein and secondly there is growing evidence that Ku could reach locally higher concentrations within bacterial cells through liquid-liquid phase separation[36]. This is where DSB repair proteins are recruited and concentrated on DSB ends to provide first protection and then accurate and efficient DSB repair which is crucial for survival and genome integrity. Interestingly, we do not observe two Ku-filaments bridging as seen in the positive stain data (B1 or B2 arrangements), which we assume is due to the DNA length constraints utilised for structural studies. Further investigations may be required to determine the exact orientation of the filaments under biological conditions.

Moreover, in stationary phase mycobacteria don't divide and it is possible that Ku plays an additional role by forming filaments with DNA, protecting and stabilising it in the coiled state, to protect the latter. This binding is potentially controlled by phosphorylation, since Ku has been shown to be phosphorylated at Thr24, Thr75, Tyr122, Ser258[37]. Thus, phosphorylation can be a mechanism regulating the DNA state depending on growth stage and environment. Interestingly, NHEJ in humans is controlled by phosphorylation by the DNA-PKcs kinase, which is not present in bacteria. However, phosphorylation may still be utilised for a way to control this critical mechanism. Further experiments are required to establish this.

The differences observed in NHEJ between humans and bacteria highlight Ku as a potential target for the development of antimicrobials. By inhibiting DNA repair in *Mtb*, the bacteria become more susceptible to host defences, facilitating efficient clearance. This study thus opens the door to leveraging this pathway as a promising therapeutic target.

## Methods
### Expression and Purification of Ku-Mtb

The bacterial expression plasmid (pET45b) coding for wild-type (WT) Ku-*Mtb* was kindly provided by Dr Eric A. Galburt. The L13A/V14A double mutant plasmid was generated by GenScript. Ku-*Mtb* or the L13A/V14A double mutant were purified according to[23] with some differences. In short, the protein was over-expressed in *E. coli* BL21 (DE3) cells at 37 °C until an OD600 reached 0.5–0.6. A concentration of

0.5 mM IPTG was used to induce the expression for 16 h at 23 °C. Cells were harvested and lysed in buffer 50 mM Tris-HCl pH 8, 0.25 M NaCl, 10% sucrose, 5 mM BME, supplemented with a protease inhibitor tablet. Lysate was sonicated and incubated with benzonase to remove DNA contaminants. The supernatant was then incubated with Ni-NTA resin equilibrated with lysis buffer and incubated for 1 h at 4 °C. The resin was washed with wash buffer (50 mM Tris-HCl pH 8, 0.25 M NaCl, 10% glycerol, 5 mM BME) and then protein eluted stepwise with wash buffers containing 50, 100-, 200-, 500- and 1000-mM imidazole. Fractions containing Ku-*Mtb* were then dialysed overnight into buffer containing low salt (50 mM Tris-HCl pH 8, 60 mM NaCl, 10% glycerol, 5 mM BME). The protein was further purified using a HiTrap Q 5 mL and eluted in high salt. Final purified protein was assessed for purity on SDS-PAGE gels and flash frozen in liquid nitrogen and stored at -80ºC for further experiments.

## DNA substrates

DNA Oligonucleotides were purchased from Sigma-Aldrich and annealed by mixing complementary strands at a 1:1 ratio. The DNA strands were heated for 5 min at 95 °C and cooled down to room temperature for at least 1 h to anneal the strands. Oligonucleotides sequences are as follows:

**FAM-DNA(Forward)**-5′ FAM-GGAATGTAACCATCGTTGGTCGG-CAGCAGGGCTTTTTTTTTTTTTTTTTTTTTT-3′

**FAM-DNA(Reverse)**-5′-GCCCTGCT GCCGACCAACGATGGTTA-CATTCC-3′

**Blunt-DNA (Forward)**-5′-CCTCTAGACCTGTAC TACTCGAGA-GATCGATCGACAGACGATGACTTAGC-3′

**Blunt-DNA (Reverse)**-5′-GCTAAGT CATCGTCTGTCGATCGATCTCGAGTAGTACAGGTCTAGAGG-3′.

## Gel mobility shift assays

Fluorescently labelled 52-mer DNA substrate (FAM-dsDNA 3′-20dT overhang, 5 nM nts) and 50 bp blunt DNA (sequences shown above) with Ku-*Mtb* protein were mixed at the indicated final concentrations in buffer containing 50 mM Tris pH7.5, 30 mM KCl, 30 mM NaCl, 5 mM MgCl2, 2 mM DTT and incubated at room temperature for 30 min. 6% Glycerol is added to the reactions prior to analysis on 6% non-denaturing polyacrylamide gel in 0.5X TBE, run at 120 V for 40 min. Gel was imaged with ChemiDoc MP (Biorad), 60 seconds exposure (Excitation source: Blue Epi, emission filter: 532/28).

## Cryo-EM sample preparation of apo Ku Mtb, Ku Mtb-DNA and the L13A/V14A double mutant

Ku-*Mtb* was concentrated using a centrifugal filter (Amicon) with a 10 kDa cut-off and buffer exchanged into 20 mM Tris-HCl pH 8, 75 mM NaCl, 5 mM MgCl2, 2 mM DTT. Ku-*Mtb* and the L13A/V14A double mutant were then mixed with FAM-DNA in a 1:4 (DNA:Ku-*Mtb*) ratio.

## Cryo-EM grid preparation

Aliquots of 3 uL of ~2.2 mg/mL of apo-Ku, WT or the L13A/V14A double mutant in complex with DNA were mixed with 8 mM CHAPSO (final concentration) to eliminate particle orientation bias (Sigma) before being applied to Holey Carbon grids (Quantifoil Cu R1.2/1.3, 300 mesh), glow discharged for 30 s at a current of 30 mA in Quorum GloQube. The grids were then blotted once with filter paper to remove any excess sample and plunge-frozen in liquid ethane using a FEI Vitrobot Mark IV (Thermo Fisher Scientific) at 4 °C and 95 % humidity.

## Cryo-EM data acquisition

All cryo-EM data was collected on a Titan Krios equipped with a Gatan K3 direct electron counting detector at the University of Leicester. All data collection parameters are given in Supplementary Table 1.

## Cryo-EM image processing

The cryo-EM processing workflow is shown in Supplementary Fig. 2 for apo Ku-*Mtb* and Supplementary Fig. 5 **for DNA-Ku-*Mtb***. The final reconstructions obtained had overall resolution which were calculated by Fourier shell correlation at 0.143 cutoff (Supplementary Table 1, Supplementary Figs. 3, 6).

## Cryo-EM structure refinement and model building

A Model was generated using modelAngelo for Ku-*Mtb*-DNA map and used as an initial template. A Model was generated using AlphaFold 2 for apo Ku-*Mtb* and used as an initial template. For Ku-*Mtb*-DNA the DNA was manually build in Coot and both models refined using Phenix real-space refinement. For apo Ku-*Mtb*, the model and the map were first run through Namdinator before refined using Phenix real-space refinement. All final refinement statistics can be found in Supplementary Table 1.

## Mass photometry

Mass photometry experiments were performed using a two^MP mass photometer (Refeyn Ltd, Oxford, UK). Data acquisitions were performed using AcquireMP (Refeyn Ltd. v1.1). Mass photometry movies were recorded with exposure times of 1 min. Silicon gaskets to hold the sample drops were fixed to a clean glass slides prior measurement. The instrument was calibrated using NativeMark Protein Standard (Thermo Fisher) prior to measurement. Prior to measurement, FAM-DNA was mixed with an increased concentration of Ku-*Mtb*, from 1:1 ratio to 1:4 ratio and incubated on ice for 10 min. The working concentration for FAM-DNA was 20 nM and for Ku-*Mtb* were 20 nM-60 nM for the actual measurement. Each sample was measured in a new gasket well. To find focus, 18 µl of Ku-*Mtb* buffer 20 mM Tris-HCl pH 8, 75 mM NaCl, 5 mM MgCl₂, 2 mM DTT was pipetted into a well. For each acquisition, 2 µl of 200 nM Ku-*Mtb*-DNA complex was added to 18 µl of buffer and measured. The data were analysed using the DiscoveMP software. The mass-to-contrast calibration curve is shown in Supplementary Fig. S2.

## FIDA 1 bio

For both experiments, the hydrodynamic radius of FAM-DNA at 45 nM was measured using 480 nm detector using Fida 1 instrument software v2.45 and predicted Rh was calculated using FIDA 1 Structural Rh predictor software. Experiments were done in buffer containing 20 mM Tris-HCl pH 8, 75 mM NaCl, 5 mM MgCl2, 2 mM DTT, using a PC coated capillary and the tray containing the sample was at 4 °C. For the first experiment, titration of Ku-*Mtb* was done by dilution in series. The working concentration for Ku-*Mtb* were 0 nM-10 µM, in a capillary mix mode (indicator and analyte were mixed in the capillary) at 10 °C. To better characterise the seconding binding a higher concentration of Ku-*Mtb* was used, 0 nM-80 uM and the complex was pre-mix prior to the measurement at 25 °C. For the first experiment, the data was fitted using the Fida software v3.0, with the first experiment being fitted using a 1:1 binding model.

## AFM DNA substrates

The same DNA oligonucleotides were used as for Cryo-EM experiments (FAM-DNA and Blunt-DNA, sequences shown above). DNA/Ku-*Mtb* complexes were prepared using FAM-DNA.

## AFM Sample preparation using divalent ion deposition

For AFM sample preparation, 0.5 cm² muscovite mica disks were freshly cleaved before deposition of DNA substrates and Ku-*Mtb* protein either separately or complexed together. DNA stocks were diluted to a working concentration of 10 ng/µL using MilliQ water. Protein stocks diluted from 215 ng/µL to a working concentration of 10.75 ng/µL using the Ku buffer (75 mM NaCl, 5 mM MgCl₂, 20 mM Tris pH 7.4). For AFM measurements of the DNA and Ku alone, 5 ng and 20 ng of molecule were deposited respectively. For samples containing DNA and protein

complexes, 5 ng DNA and 20 ng protein were complexed together at a 1:4 DNA:Ku-*Mtb* molar ratio in an Eppendorf for 10 min at room temperature before depositing onto the mica. For deposition on the mica, a 20 μL droplet of Ku buffer (75 mM NaCl, 5 mM MgCl2, 20 mM Tris pH 7.4) was added followed by the DNA, Ku-*Mtb* or the DNA/Ku-*Mtb* mixture. The droplet was then mixed rigorously using a micropipette. The sample was left to adsorb for 10 min at room temperature before rinsing 4 times with imaging buffer (3 mM NiCl2, 20 mM HEPES pH 7.4). AFM imaging in liquid was performed in a droplet containing 20 μL of fresh imaging buffer on the mica surface.

## AFM using APS deposition
An alternative deposition method was also used to exclude divalent ions from the buffers. In this case the mica surface was modified by adding 20 μL of 0.17 mM APS (aminopropyl silatrane, Sigma Aldrich), allowing functionalisation for 30 min at room temperature. The mica was then washed with MilliQ and gently air dried with compressed air. 20 μL of divalent ion-free Ku buffer (75 mM NaCl, 20 mM Tris pH 7.4) was added to the APS-functionalised mica and DNA, protein or DNA/Ku-*Mtb* complexes prepared as above were left to adsorb for 30 min at room temperature before rinsing 4 times with divalent ion free Ku buffer (75 mM NaCl, 20 mM Tris pH 7.4). AFM imaging in liquid was performed in a droplet containing 20 μL of fresh divalent ion free imaging buffer on the mica surface.

## AFM imaging in liquid
All AFM measurements were performed using a commercially available AFM (Dimension XR FastScan, Bruker) in PeakForce™ Tapping at room temperature in aqueous environments. Standard, soft, in-liquid AFM cantilevers (FastScan-D, Bruker) with a nominal tip radius of 5 nm and a spring constant of 0.25 N/m were used. Forces applied to the sample ranged from 70 to 300 pN and PeakForce amplitude ranged from 10-18 nm at a PeakForce Tapping frequency at 8 kHz. Images were recorded with a minimum resolution of 1.5 nm per pixel.

## AFM image processing
All raw AFM images were processed with open-source AFM image analysis software TopoStats[38]. Pre-processing is required to remove standard AFM image artefacts before generating statistics on imaged molecules (grains) using a 'config.yaml' file that allows for preprocessing parameters to be configured. First, raw images were first flattened by both line and plane flattening. Grains were then masked using a configurable height threshold to differentiate the grains from the background before flattening the background by further line flattening. The height distribution was adjusted to ensure that the image's background is 0 nm and a 1.1px Gaussian filter was applied to reduce high-frequency noise.

The configuration file was used to generate statistics on grains above a threshold of 1 nm in height and between an area of 10 and 20,000 nm, whilst excluding small grains/noise below 15 nm², to mask the grains of interest i.e. DNA, proteins or DNA-protein complexes. These masked grain areas were then used to calculate the smallest bounding length, volume, max Feret, mean heights and max heights of the grains. The images were produced using varying Z-range values as stated in images and captions to maximise visibility of grain features.

## AFM plotting and statistical analysis
The generated statistics were plotted as a strip plot, with a KDE or box plot overlay, using Seaborn and Matplotlib packages in a Python Jupyter Notebook. The significance of the differences in DNA substrates' smallest bounding length and the differences in volume of FAM-DNA or KuMtb in comparison to FAM-DNA with KuMtb were accessed using Mann-Whitney tests and differences between FAM-DNA, KuMtb and FAM-DNA with KuMtb volume were accessed using Krustal-Wallis tests, followed by Dunn's test with Holm correction.

## EM positive staining experiments
EM Positive staining method and dark-field imaging mode were used to analyze Ku-*Mtb*- DNA complexes. Hexagonal 600 mesh copper grids previously covered with a thin carbon film were functionalized in a homemade device by glow-discharge in the presence of amylamine, providing NH$_3^+$ charges deposition onto the carbon surface[39,40].

DNA fragments (1 nM) were mixed with 50–200 nM of Ku-*Mtb* in binding buffer (10 mM Tris-HCl pH 8, 50 mM NaCl, 5 mM Mg$^{2+}$) for 30 min at 37 °C. 5 μl of sample were deposited on the activated carbon film grids for 1 min and then rinsed with aqueous 2% (w/v) uranyl acetate and dried to simultaneously spread and stain the DNA complexes onto the surface. The samples were observed in dark field imaging mode and the contrast is optimized in zero loss filtered conditions using a Zeiss 912AB microscope equipped with an omega filter. The images were captured with a Tengra CCD camera at magnifications from 28980 to 50809 and analyzed by Item software (both Olympus, Soft Imaging Solutions).

**DNA substrates.** The blunt substrates were amplified from the pBR322 plasmid (NEB) by PCR using *Taq* polymerase (NEB) and the pairs of primers (Supplementary Table 1). For each 3' overhang substrates, 2 DNA fragments of 401, 404, 407 and 443 bp were amplified from the pBR322 plasmid by PCR using a pair of primers among one is biotinylated (Supplementary Table 1). Biotinylated PCR products were purified by chromatography using a MiniQ 4.6/50 anion exchange column (Cytiva). For each DNA construction the 2 ssDNA were obtained.

For one DNA construction, the appropriate couple of biotinylated dsDNA were loaded onto a HiTrap Streptavidin HP column (Cytiva) in equimolar concentrations. Purification of the non-biotinylated single strand DNA was achieved by elution with 60 mM NaOH. The ss-dsDNA construction results from annealing of ssDNA optimized from 95 °C (3 min) to room temperature[41].

## Cultivation of *M. smegmatis*
*M. smegmatis* mc²155 was grown in 7H9 Middlebrook broth supplemented with 0.2% (v/v) glycerol, 10% (v/v) albumin-dextrose-catalase (ADC), and 0.05% (w/v) Tween 80 (hereafter 7H9 medium) at 37 °C with shaking at 200 rpm. CFU counts were determined on 7H10 Middlebrook agar supplemented with 0.2% (v/v) glycerol and 10% (v/v) albumin-dextrose-catalase (ADC). When required antibiotics and chemicals were added at the following concentrations: kanamycin - 50 μg/ml, hygromycin - 50 μg/ml, X-Gal - 50 μg ml$^{-1}$, sucrose - 2% (w/v), MMS – 0.1 % (v/v).

## Generation of Δku mutant and complemented mutant
The ku unmarked deletion mutant was generated using homologous recombination method a previously described[42]. Briefly, ~1 kb flanking regions were amplified from *M. smegmatis* genome using Platinum Taq polymerase (ThermoFisher) and primers 5580FR1F 5′-AacAGTACTcactggttggatcggtgc-3′, 5580FR1R 5′-catAAGCTTagtatggcgtaccgcacggtt-3′, 5580FR2F 5′-catAAGCTTgccaagaaggcagctgcgaa-3′, 5580 FR2R 5′-ttcGTCGACcaggtcgagtgttgcgacgt-3′ and cloned in ScaI and HindIII, HindIII and SalI sites of suicide plasmid p2NIL. This was followed by cloning of PacI cassette containing markers for single-crossover strains. The construct was electroporated in *M. smegmatis* and plated on 7H10 agar containing kanamycin and X-Gal. Blue colonies were grown without kanamycin to allow double cross-over and mutants were selected on 7H10 plates containing sucrose. Deletion mutant candidates were confirmed by PCR using primers 5580testF1 5′-gaacgtgttccaccgcac-3′ and 5580testR 5′-atccggcagaacgccga-3′ and whole genome sequencing (SeqCentre). For complementation msmeg_5580 coding sequence and upstream region containing a putative promoter was amplified from *M. smegmatis* genome, using Platinum Taq polymerase and primers 5580com F

5'-actGAATTCgttccaccgcacaaacag-3' and 5580comR 5'-agaAAGCTTc-tacgacttcttcgcagctgc-3'. The product was cloned in EcoRI and HindIII sites of pMV306, an integrating mycobacterial vector. L23A/V24A mutation was done using Platinum™ SuperFi ™ DNA polymerase and primers kuL23AV24AF 5'-ccttcggcGCggCGaacgtgccggtcaagg-3' and kuL23AV24AR 5'-GGCACGTTcgCCgcGCCGAAGGCGATCGAA-3'.

## Phenotypic studies

*M. smegmatis* strains were incubated in 7H9 medium for 7 days (stationary phase) before exposure to MMS or desiccation experiments. MMS exposure: *M. smegmatis* cultures were diluted to OD580 = 0.1 in 7H9 medium, serially diluted and plated on 7H10 agar (for CFU counts) before addition of MMS and incubation at 37 °C with shaking at 200 rpm for 6 h.

Desiccation−stationary phase *M. smegmatis* cultures were centrifuged at 4000 x $g$ for 10 min, pellets were resuspended in 10 mM MgSO4 followed by centrifugation at 4000 × $g$ for 10 min. Resultant pellets were resuspended in 10 mM MgSO4. For each replicate 10 2 µl drops were spotted in one well of 24 well tissue plate and dried for 2 h in safety cabinet and further incubated in the dark at room temperature for 24 h. For CFU counting dry pellets were resuspended in 200 µl 7H9 medium, incubated for 1 h without shaking, serially diluted and plated on 7H10 agar. Unpaired $t$ test or one-way ANOVA (Prism 10) were used to evaluate the statistical differences in the survival patterns.

## Reporting summary

Further information on research design is available in the Nature Portfolio Reporting Summary linked to this article.

## Data availability

All structural data presented are publicly available. Cryo-EM structures and maps are deposited at the PDB and EMDB with accession codes as follows: 9IG8 and 9I91, EMD-52849 and EMD-52748. Source data are provided with this paper.

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

## Acknowledgements

We acknowledge the Midlands Regional Cryo-EM Facility at the Leicester Institute of Structural and Chemical Biology (LISCB) and major funding from MRC (MC_PC_17136). We thank the Lister Institute of Preventative Medicine Prize for support of this research. We would also like to thank the Medical Research Council for the two standard research grants (MR/X00029X/1 and UKRI699). A.P. and H.R. wish to acknowledge UKRI funding in the form of a Future Leaders Fellowship (MR/W00738X/1) to A.L.B.P and EPSRC funding via the Henry Royce Institute for Advanced Materials (EP/R00661X/1, EP/S019367/1, EP/P02470X/1 and EP/P025285/1) and Xinyue Chen for Dimension FastScan access and support through Royce@Sheffield. F.W. acknowledges funding from the European Research Council (consolidator grant, no. 866238) and the Swedish Research Council (2020–03400). GM was funded by the National Institute for Health and Care Research (NIHR) Leicester Biomedical Research Centre grant number NIHR203327. LB work was supported by BBSRC and University of Leicester funded Midlands Integrative Biosciences Training Partnership (MIBTP), grant number BB/T00746X/1. E.G. thanks NIH R35GM144282.

## Author contributions

A.K.C. directed the study and led the experimental design. S.Z. and S.A. expressed and purified the Ku and performed biophysical experiments and cryo-EM data collection, processing and analysis. S.B and E.L.C. conducted positive stain imaging experiments and analysis. H.S., H.R. and A.P. carried out AFM experiments and analysis. L.B. and G.M. generated mycobacterial mutants and performed mycobacterial survival experiments and analysis. F.M. and F.W. carried out EMSA studies and biophysical analysis. T.B., E.G.P.S., and J.W. advised, supported and helped with FIDA experimental design and analysis. S.W.H. helped with cryo-EM data processing, analysis and advice. A.C. and E.G. provided the initial Ku plasmid and protein, project discussions, and advice. All authors edited the manuscript. For the purpose of open access, the author has applied a Creative Commons Attribution license (CC BY) to any Author Accepted Manuscript version arising from this submission.

## Competing interests

The authors declare no competing interests.
