## [Transparent Peer Review file · Nature Communications]

Oligomerisation of Ku from *Mycobacterium tuberculosis* promotes DNA synapsis

Corresponding Author: Dr Amanda Chaplin

Version 0:

Reviewer comments:

Reviewer #1

(Remarks to the Author)

Zahid et al., report the exciting cryo-EM structure of *Mycobacterium tuberculosis* Ku on its own and bound to DNA. Interestingly, they find that Ku forms filaments along the DNA. They propose that in bacterial NHEJ, which lacks DNA-PKcs and other proteins that synapse the DNA double-strand break, Ku filaments are needed to form the stable synapse. The cryo-EM structures also show that the Ku C-terminus is mobile – packed against apo Ku and not visible in the DNA-bound Ku, indicating it has been displaced, potentially in a way to allow for interaction with LigD. Supporting data from AFM, EMSAs, mass photometry, FIDA and positive staining EM confirm Ku forms higher order oligomers when bound to DNA and visualizes previously reported Ku-DNA bridging activity.

This is the first reported structure of the bacterial Ku protein from NHEJ, which will be of interest to those in the DNA repair field, and which provides interesting insights on how bacterial Ku has more functionality than eukaryotic Ku in NHEJ. Previously reported work on bacterial NHEJ has shown biochemically and through single-molecule studies that Ku bridges and synapses the DNA ends. This cryo-EM structure, along with AFM and EM, is the first to visualize it and provide molecular details about the event. This builds on the bacterial NHEJ mechanism and explains how in the absence of proteins found in eukaryotic NHEJ that bridge and synapse the DNA ends, bacterial NHEJ only needs Ku for this DNA-binding activity.

Overall, the work is well done. However, the AFM data needs some further detail. Heights and volumes of DNA, Ku and Ku-DNA complexes are reported, but few images are shown and the number of particles measured to get these values are not reported. What was the distribution of particle volumes and heights, to ensure that Ku alone did not form larger oligomers in the absence of DNA? Of minor note – in the Extended Data Figure 8, it says the single Ku (1) and Ku aggregate (2) are 1260 and 4700 nm³, respectively, but in the image and height trace, it appears that the particle labeled as 1 and masked in pink is larger – this may be a mislabel in the figure?

Also, there is only a single image of the Ku-DNA complex shown (Figure 2D). Did all the Ku-DNA complexes appear as similar large aggregates and what kind of distribution was observed in the heights and volumes? Was more than one large aggregate observed and/or were there other smaller Ku-DNA complexes? This would help answer whether the large Ku-DNA aggregate occurs frequently or rarely. Having this data would strengthen the observations and conclusions.

There are a few minor details to address that may also benefit the publication. These are listed below:

1. In the introduction, lines 63-64, it is important to note that while only structures of Mtb ligase and polymerase domains have been solved by x-ray crystallography, there are crystal structures of the LigD phosphoesterase domain as well, but from *P. aeruginosa* (PDB 3N9B).
2. Figure 1A,B – The bars displaying p-value are only between the wildtype Ku strain and the Δ ku pMV306 strain, but not Δ ku. It looks like there should be a significant difference between wildtype and Δ ku as well, but if not, why?
3. In the Extended Data Figure 3, the pLDDT scores, PAE, ipTM and pTM scores should be reported for the predicted models, especially as the AlphaFold2 model for Ku was used as an initial template for the cryo-EM model building.
4. It would be interesting to see how well apo and DNA-bound Ku align. Is it only the C-terminus that moves, or do other structural features change as well?
5. Lines 291-293 state that the loop interface is more critical than the helix interface in Fig 3D. It is unclear from the written text why this conclusion is made and how the data support this conclusion.

Reviewer #2

(Remarks to the Author)

“Oligomerization of Ku from Mycobacterium Tuberculosis DNA synapsis.” by S. Zahid et al. presents the biochemical and structural characterization of half the mycobacterium’s two component non-homologous end joining pathway. The key thesis of the proposed research is the pathogenicity of Mtb is related to the genomic integrity maintenance associated with the NHEJ pathway. Therefore, the characterization biochemically and structurally of the Mtb Ku homodimer and its DNA binding will be a significant step in uncovering mechanistic processes that could be targeted for inhibition. The work presented supports this hypothesis by first presenting the role Ku plays in surviving DNA damage and desiccation stress, albeit not essential nor a new finding. Further, this point could benefit from references to supporting research in Mycobacterial NHEJ pathway affecting antibiotic resistance or infectivity. The presentation of the Mtb apo-Ku cryoEM structure demonstrates the similarity between the core protein fold of the heterodimeric Ku70/80 from eukaryotes and the homodimeric bacterial Mtb Ku even though the small size (60 kDa) of the Ku homodimer limited the overall attainable resolution to 4.06 Å. This could likely be improved with lower KeV data collection, (200 KeV versus 300 KeV) to improve the SNR in the particle images leading to better parameter fitting. However, the high similarity of the Alpha-fold structures to the relatively good resolution cryoTEM structure suggests effort here would be of limited return.

When Ku is bound to a 32 bp dsDNA with a 20 nucleotide poly-T 3’ overhang, Ku formed a filamentous structure that enabled cryoEM structure determination to 2.96 Å which was suitable for accurate placement of individual homodimers on dsDNA. Dimer to Dimer interfaces are well resolved in these maps both between strand Ku homodimers and interstrand bound Ku homodimers. AFM experiments were also presented but don’t really add any clarity to the assembly as they show poor distribution of the filamentous complex and only show that a consistent height on 8 nm is conserved from Ku alone to Ku-DNA filament. Further characterization of the Ku DNA filament was carried out by mass photometry and flow induce dispersion analysis which confirmed many of the conclusions of the cryoTEM derived filament structure while providing dissociation and assembly kinetics. A brief description of the filament Ku formation was provided which details the association is driven by hydrophobic helical and loop residues being buried between the dimer interfaces. This would likely generate a highly variable surface that may explain why the filament appears highly variable in the reconstruction process. The structure does not identify any large conformational changes occur from apo-Ku, with the single exception of the c-terminal extended domain which is not resolved but instead lost in the structure when bound to DNA. Presumably, this absence of density is due to displacement similar to the single stranded region of the oligo nucleotide used to build the Ku filament. Finally, larger DNA fragments were bound to Ku and examined by positive staining TEM. Imaging showed recircularization events that varied in end joining topology with the authors stating greater than 80% recircularization by Ku occurred, this data was further quantified in extended data table 2, where a range of 3’ ends were examined.

Overall, the paper is concise and coherent adding structure information to the bacterial homologue of eukaryotic Ku70/80. It reconfirms many details of bacterial Ku biochemistry already in the literature for *B. subtilis* and *M. smegmatis* while adding the apo-Ku as well as Ku bound to DNA structures that are at higher resolution, but not significantly more informative <https://doi.org/10.1093/nar/gkac906>. As an NHEJ enthusiast, inclusion of the LigD in the structure determination and biochemical analysis would make for a more comprehensive treatise, minimally deletion of the c-terminal extended domain of Ku to demonstrate their proposed recruitment mechanism for LigD either via rescue of desiccation or MMS sensitivity analysis could easily be performed. The loss of the ssDNA, although easily passed off as being due to the highly flexible nature of ssDNA or maybe a contaminate carried over in the purification thus degrading the ssDNA region raises the question of whether the structure best resolves the blunt end of the oligonucleotide used for assembly or the single strand overhang end. If it is degraded, the resolution of the map is insufficient to determine pyrimidine versus purine on the individual strands to properly fit the sequence, so it is not possible to know. Is the ssDNA region even present in the Ku DNA assembly given the amount of other bands in the purification Mtb Ku (supplementary Fig. 1)? The cryoEM structure of the blunt ended DNA topology bound to Ku was also solved but if the ssDNA is truly missing this would suggest the structures are essentially the same. Unfortunately, the data in extended data 7 presenting the additional structure appears to not have a PDB or EMDB accession or just wasn’t provided to the reviewers. DNA end topology (blunt and single stranded were examined but not stem loop or blocked ends) effects on Ku binding and fiber formation would be an interesting addition to this work as the delegation of HR versus NHEJ is growth phase dependent suggesting certain structures may be recognized preferentially. Indeed, the positive staining TEM experiments suggest differences arise in fiber assembly based but bringing this to the cryoEM structural level of insight or more rigorous biochemical level would be of interest. As the manuscript stands, it is abbreviated in its significance to the field but could be improved with exploring further details such as the c-terminal extended domains relevance, possibly exploring the LigD interaction structurally even if only the portion of LigD that interacts with Ku were used, and possibly seeing if focused refinement or variability analysis across the filament helps resolve the more interesting details in the fiber formation. The findings feel superficial and not explored fully given the potential presented within the paper and the previous work in the field.

Minor Points:

- 1.) Model Angelo results need to be validated as the Ku fiber models have lysine residues that have the atomic coordinates of glycine, those found are position 117 and 184 in KuDNAModel.pdb chains ABEF.
- 2.) Extended figure numbering in the main body text is wrong. Line 186 Extended Figure S7 should refer to S8 and again on line 325 Extended Figure S8 should be S9.
- 3.) Line536...FAM-DNA (Reverse) at the 3’ end read CCTACA but the complementary strand and structure suggest it should read 3’-CCTTACA-5’

Reviewer #3

(Remarks to the Author)

Zahid et al. report cryo-EM structures of mycobacterial Ku with and without DNA bound in 4.04 Å and 2.96 Å resolution, respectively. They furthermore show Ku is important for mycobacterial survival under DNA-damaging conditions. The structures reveal that an interesting C-terminal blocking helix prevents filament formation in DNA-free form. Upon DNA binding, the blocking helix is replaced by another Ku heterodimer to form a Ku filament. The DNA end gaps distance spans ~ 40 Å, representing a synaptic complex before LigD repairs breaks. The authors also showed in positive-stained EM that Ku can circularize DNAs and hold DNA ends in proximity. These results are indeed unique in the bacterial NHEJ repair system, and the structures reported show how this works. Readers will also appreciate that authors made several biophysical measurements to understand the biophysical properties of filament formation and stoichiometry.

Ku structural information remains limited, and no structures are available for Ku complexes from any bacterial species. The newly determined DNA-free and bound cryo-EM structures of Ku are an important contribution. The data and insights furthermore have potential for informing bacterial Ku biology and possibly targeting. Before publication the authors should address the following points.

Points.

1. Why does Ku-Mtb need to form filaments along DNA ends? This was not clearly discussed in the text. Human NHEJ proteins can also form filaments (e.g. Hammel et al XRCC4 protein interactions with XRCC4-like factor (XLF) create an extended grooved scaffold for DNA ligation and double strand break repair. *J Biol Chem.* 2011 Sep 16;286(37):32638-50. doi: 10.1074/jbc.M111.272641; Mahaney et al., XRCC4 and XLF form long helical protein filaments suitable for DNA end protection and alignment to facilitate DNA double strand break repair. *Biochem Cell Biol.* 2013 Feb;91(1):31-41. doi: 10.1139/bcb-2012-0058). Are there relationships or functional analogies here? Readers may wish to relate these results to what is known about the human system.
2. Have the authors observed any two Ku-filaments bridging interactions in cryo-EM that mimic the circular end bridging (B1 or B2) in positive stained EM?
3. What are the critical structural differences between KuMt and human Ku70/80 that human Ku cannot hold broken DNA ends? The overlay in Fig. S5 is unclear. It should be improved for clarity. One key element is the filament interface – is this similar or not? If not, what is blocking the filament formation in human Ku70/80?
4. While Ku filament formation is a novel observation in bacterial Ku, one of the purposes of understanding Ku-Mtb structures is for therapeutic potential. Can the authors design interface mutants that prevent filament formation and determine whether the interface mutants are critical for mycobacterial survival under damaging agents/stress?
5. Ku filament formation along DNA ends seems both intriguing and puzzling. For long DNA fragments, how does Ku form filaments along the DNA ends, given that the DNA binding affinity is much tighter than filament formation? Does Ku homodimer slide down the DNA ends, or does the DNA binding channel of Ku open to bind DNA? Are the authors aware of any examples that show Ku can open the DNA binding channel? If not, it seems likely that the sliding event happens. The authors show circularization is independent of DNA end overhang length (Extended Data Table 2), while Robin Oz et al. show 4 nt complementary DNA overhangs increase synapsis lifetime drastically (*Nucleic Acids Research* 49, 5, 18 March 2021, 2629–2641, <https://doi.org/10.1093/nar/gkab083>). What are authors' views on how Ku forms a filament along DNA ends? This should be concisely noted.
6. Do the authors see any structure differences between the DNA gap interface and the DNA engaged interface?
7. Line 185 should be Extended Data Fig S8. Not S7.
8. For readers to follow a line of logic, the figures should be cited in order instead of randomly.

Reviewer #4

(Remarks to the Author)

Zahid et al. reported cryo-EM structures of mycobacterial Ku with and without DNA bound in 4.04 Å and 2.96 Å resolution, respectively. They also showed Ku is important for mycobacterial survival under DNA-damaging conditions. The structures revealed a C-terminal blocking helix prevents filament formation in DNA-free form. Upon DNA binding, a blocking helix is replaced by another Ku heterodimer to form a Ku filament. The DNA end gaps distance ~ 40 Å, representing a synaptic complex before LigD repairing the breaks. They also showed in positive-stained EM that Ku can circularize DNAs and hold DNA ends in proximity. This is indeed unique in the bacterial NHEJ repair system, and the structures here show how. I also appreciate that authors made several biophysical measurements to understand the biophysical properties of filament formation and stoichiometry.

Specific comments.

1. Why does Ku-Mtb need to form filaments along DNA ends? It was not clearly discussed in the text.
2. Have authors observed two Ku-filaments bridging interaction in cryoEM mimicking the circular end bridging (B1 or B2) in positive stained EM?
3. What are the critical structural differences between KuMt and human Ku70/80 that human Ku cannot hold broken DNA

ends? The overlay in Fig. S5 is unclear and needs improvement for clarity. One key element is the filament interface, similar or not? If not, what is blocking the filament formation in human Ku70/80?

4. While Ku filament formation is a novel observation in bacterial Ku, one of the purposes of understanding Ku-Mtb structures is for therapeutic potential. Can authors design interface mutants that prevent filament formation and determine whether the interface mutants are critical for mycobacterial survival under damaging agents/stress?

5. It is intriguing and puzzling about Ku filament formation along DNA ends. For long DNA fragments, how does Ku form filaments along the DNA ends, given that the DNA binding affinity is much tighter than filament formation? Does Ku homodimer slide down the DNA ends, or does the DNA binding channel of Ku open to bind DNA? I haven't seen any examples that show Ku has the ability to open the DNA binding channel. So it is likely the slide event happens. The authors show circularization is independent of DNA end overhang length (Extended Data Table 2), while Robin Oz et al. show 4 nt complementary DNA overhangs increase synapsis lifetime drastically (Nucleic Acids Research, Volume 49, Issue 5, 18 March 2021, Pages 2629–2641, <https://doi.org/10.1093/nar/gkab083>). What are authors' opinions on how Ku forms a filament along DNA ends?

6. Do authors see the structure difference between the DNA gap interface and the DNA engaged interface?

7. Line 185 should be Extended Data Fig S8. Not S7. And it would be great if the figures could be followed in order instead of randomly.

Reviewer #5

(Remarks to the Author)

The authors present the results of extensive structural studies of complexes with DNA Ku protein of *Mycobacterium tuberculosis* bacteria. Similar to Ku protein of vertebrates, the bacterial Ku protein is critically needed for the non-homologous end-joining (NHEJ) DNA repair, however process can be different as multiple proteins are involved in the NHEJ process for vertebrates. The authors applied multiple experimental approaches to characterize the complexes of Ku protein with DNA substrates and found the oligomerization of the protein bound to DNA is the key factor requiring for the synapsis of the DNA ends. CryoEM data led the authors to the model of the end-joining with the key role of the C-terminus of Ku in this process. However, a number of issues need to be clarified to make a convincing case.

1. The authors used a hybrid DNA substrate with 32 bp DNA duplex tailed with 20 nt ssDNA in the majority of their studies including high-resolution cryo-EM studies. However, there is no indication in the CryoEM data to the ability of Ku protein binding ssDNA. Only two Ku dimers per 32 bp DNA duplex. What is the role of ssDNA tail in the complex assembly? It not identified in the cryo-EM data.

2. AFM data are not in line with EM data including CryoEM ones. First, picture in Fig 2D shows a particle with curved Y-shape, whereas the Cryo EM data in Fig 2C show linear filaments. If the images of complexes are different and in-line with cryo-EM data, these should be presented, and the population of various assemblies should be characterized.

3. AFM volume measurements – they suggest that almost 100 Ku protein are bound. Apparently more than one DNA template was in that assembly. Are these typical data? Was the sample the same as used for CryoEM studies? Again only linear filaments are shown in Fig. 2C – did such filaments seen in the AFM data?

4. The height measurements in Fig 2D – only one cross-section is shown – what are other measurements? The analysis over the statistically significant set of images to support the conclusion is missing.

5. The model of the Ku-DNA complexes suggests a coating of the DNA duplex with the protein with a high density – two dimers per three DNA turns. Why only a limited DNA segment is coated with proteins with experiments with long DNA substrates (Fig. 4) with a high concentration of the protein?

6. DNA cyclization by Ku protein. According to Fig. 4, two types of circles are formed. Apparently only one of them is an appropriate substrate for LigD ligase. What is the ratio between the two types of DNA rings depending on the Ku protein concentration? Could it be that the right type of rings is formed at lower concentration of the protein? What would be the Ku stoichiometry for the rings formed at lower concentrations of Ku? Experiments with lower Ku concentration are needed to clarify this physiologically important property of Ku.

Reviewer #6

(Remarks to the Author)

Reviewer #7

(Remarks to the Author)

This manuscript presents a structural and functional analysis of Ku from *Mycobacterium tuberculosis* (Ku-Mtb). They demonstrate its ability to form oligomers on DNA and support DNA end synapsis during non-homologous end joining (NHEJ). The authors combine cryo-EM, mass photometry, AFM, positive stain EM, and biochemical assays to reveal that Ku-Mtb filaments coat DNA and facilitate recruitment of LigD for ligation, bypassing the need for DNA-PKcs present in eukaryotic NHEJ systems. Importantly, they identify the displacement of a C-terminal α -helix during filament formation and propose a model where this movement promotes LigD recruitment.

This work is original, comprehensive, and timely, as it sheds light on bacterial NHEJ mechanisms. The structural insights provided are novel, particularly in highlighting filament formation by Ku-Mtb—a behavior not observed in human Ku70/80.

The study is of broad interest to structural biologists, microbiologists, and researchers in DNA repair. Overall, the EM, EMSA and mass photometry data appear to be of high quality; however the AFM data shown in the paper is minimal, although there is discussion of population analysis, no data are shown, and perhaps more concerning the AFM experiments were conducted at extremely low salt, bringing into question the physiological relevance of these experiments.

Major Issues:

1. AFM Experimental Limitations (Data Presentation, Replication, Controls)

o The AFM analysis is a critical part of the study but lacks essential detail and rigor. The authors show height profiles from only a single trace per condition without reporting the number of molecules analyzed (N), number of replicates, or any statistical variation (e.g., standard deviation or SEM). No histograms or frequency distributions of volume measurements are provided, making it difficult to assess reproducibility or heterogeneity. The authors should present distributions and include the number of complexes analyzed.

o The authors show only a single complex which appears relatively large. This large complex would not be expected based on their EMSA studies. It is likely that the large complexes are an artifact of the very low salt and high divalent ion concentration (25mM MgCl₂, 10mM Tris pH 7.4). It is likely that the authors are using these conditions to keep the complexes attached for solution imaging; however there are other choices (e.g., poly-L-lysine, APTES, or even native Mg²⁺ adsorption) that allow physiological salt concentrations to be used. It is unclear why the authors chose solution imaging, which is much more difficult than air imaging, when the data the authors desire could be easily obtained using air imaging. The authors underutilize the potential of in-liquid AFM. For example, they could support their filament model with dynamic imaging (movies), conformational change analysis, or bend angle quantification, which are all feasible and would strengthen their claims about higher-order oligomeric states and synapsis.

o There is no mention of appropriate AFM controls. Without showing DNA-only and Ku-only traces in parallel under matched imaging conditions and buffer, it is hard to distinguish specific Ku-DNA filament formation from nonspecific aggregation.

2. Positive stain EM classification criteria

o Similar to the AFM studies, the authors do not show distributions of species and do not give the number of complexes analyzed. This analysis would benefit from a length distribution graph for the linear and circular structures

3. C-terminal α -helix movement and LigD recruitment (interpretation clarity)

o While the proposed model for C-terminal α -helix displacement and LigD recruitment is intriguing, it is speculative. No direct evidence for LigD interaction with Ku-Mtb in this study is shown. The authors should clarify what is directly supported by cryo-EM versus inferred from prior work. Indicate that LigD binding remains to be validated experimentally.

Minor Issues:

1. The numbering of extended figures is inconsistent throughout the text (e.g., Extended Data Fig. 7 is actually Fig. 8).

2. Figure 5 (model) could be improved by clearly distinguishing experimentally validated structures from modeled components (e.g., LigD).

3. Textual clarity would benefit from paragraph spacing in the Discussion to ease readability.

4. Clarify if Ku-Mtb expression levels (e.g., stationary phase) were quantified or inferred.

5. Minor typos (e.g., "Cterminal" instead of "C-terminal" in figure legends) should be corrected.

Version 1:

Reviewer comments:

Reviewer #1

(Remarks to the Author)

The authors have addressed all previous reviewer comments from the initial review successfully. Two minor points to consider are included below when making final edits for publication, otherwise, the article is highly significant for the field, with the work supporting the conclusions drawn and has used appropriate methodology to achieve these results.

Minor points:

1. For supplementary figure 7, please include an RMSD for the aligned structures to provide a quantitative measure.

2. On page 10, line 261, nm should be nm³.

Reviewer #2

(Remarks to the Author)

"Oligomerization of Ku from Mycobacterium tuberculosis DNA synapsis." by S. Zahid et al. has improved the manuscript significantly.

Of interest is the improved statistics presented for the AFM and positive stain DNA topology experiments.

In support of their hypothesis for the c-terminal helices role in filament or synapse formation, mutations to two residues L13A and V14A proximal the helices but within the dimer interface were generated that disrupted filament formation as assessed cryoTEM structure analysis. The c-terminal protomer helices still look to be displaced in the DNA bound KuMTB mutant cryoTEM structure, but is not discussed. On line 312 of the revised manuscript the authors' state, "Mutations introduced at the helical interface did not disrupt oligomer formation." The nature of these mutation attempts were not detailed in the manuscript and are still useful information regardless of the negative outcomes, additionally how it was determined these mutations failed to disrupt filament formation would also be useful information.

Additionally, the authors directly address DNA end configuration and filament formation and KuMTB binding. Together with the mutational analysis, this is a significant addition to the understanding of the filament formation process by the MTB Ku

DNA complex.

Due to these revision, the work is sufficiently improved and has significant relevance to multiple fields to warrant consideration for publication.

Reviewer #3

(Remarks to the Author)

The revised manuscript is suitable for publication. However, one added modification is needed to the sentence "Invertebrates including, bacteria and yeast do not contain DNA-PKcs-like enzymes meaning an enormous number of organisms (more than the number of vertebrate organisms) carry out NHEJ without DNA-PKcs." This statement should be updated in the light of the 2021 publication noting "...DNA-PKcs is widely distributed in invertebrates, fungi, plants, and protists..." and this publication should be cited (Lees-Miller JP, et al., Uncovering DNA-PKcs ancient phylogeny, unique sequence motifs and insights for human disease. *Prog Biophys Mol Biol.* 2021 Aug;163:87-108. doi: 10.1016/j.pbiomolbio.2020.09.010).

This revised manuscript is a significant contribution to the field, and will be of general interest. The authors created new interface mutants (L13A/V14A) and have found the double mutant disrupted the Ku-Mtb filament formation and ability to bind synaptic DNA ends, as well as decreased cell survival under DNA-damaging conditions. They also added new figures to improve the clarity to support their model and conclusion.

The implications of these findings could extend to understanding how similar mechanisms operate in other organisms, such as humans. Additionally, further exploration of the mutant's effects could pave the way for novel therapeutic strategies targeting DNA repair pathways in *Mycobacterium tuberculosis*.

Reviewer #4

(Remarks to the Author)

The authors have addressed all my concerns in their rebuttal. They created new interface mutants (L13A/V14A) and have found the double mutant disrupted the Ku-Mtb filament formation and ability to bind synaptic DNA ends, as well as decreased cell survival under DNA-damaging conditions. They also added new figures to improve the clarity to support their model and conclusion. This is a significant contribution to the field, and it would be great to see this paper published. The implications of these findings could extend to understanding how similar mechanisms operate in other organisms, such as humans. Additionally, further exploration of the mutant's effects could pave the way for novel therapeutic strategies targeting DNA repair pathways in *Mycobacterium tuberculosis*.

Reviewer #5

(Remarks to the Author)

The authors' responses are not satisfactory. I keep my comments in the same order as originally.

1. The use of the hybrid DNA template with 32 bp DNA duplex tailed with 20 nt ssDNA. The KU-mediated template assembly with such an asymmetric monomer will lead to three types of the dimers: blunt end-DNA duplex- ssDNA+ssDNA-DNA duplex-blunt end; blunt end-DNA duplex- ssDNA+blunt end-DNA duplex- ssDNA; ssDNA-DNA duplex-blunt end+ blunt end-DNA duplex- ssDNA. However, only the dimer with the structure (blunt end-DNA duplex-blunt end+ blunt end-DNA duplex-blunt end) is presented as a model based on structural studies with cryoEM. However, no such a template with blunt ends is used. Although a visualization of ssDNA tail with cryoEM can be a problem, the tail is large and can contribute to the monomer's assembly in the Ku-DNA arrays. Moreover, the 4 nm distance between the DNA ends in the model is close to the size of ssDNA coil with 20 residues. If there is no Ku binding to these oligoT ends, the ssDNA segments can work as spacers between the DNA templates. Additionally, assembly of dimers and higher oligomers via ssDNA tails can lead to branched structures observed in the cryoEM images. Unfortunately, this fundamental problem is not discussed in the paper nor addressed by the authors to my original concern. Given that the major function of Ku protein is to join the DNA ends, primarily with the blunt end morphology, the experiments with DNA template with blunt ends are required to validate/test the model proposed by the authors.

2. CryoEM vs AFM data. These are fundamentally different images. Large complexes with no clear linear arrays appear in all AFM images in the manuscript. The difference between the CryoEM vs AFM results is apparently due to the differences in the sample preparation procedures. In AFM, the samples are prepared at the liquid-surface interface, which can contribute to the higher order assembly of the DNA-protein complexes. The use of the DNA template with ssDNA tail can be an additional factor contributing to clustering observed with AFM. Importantly, even Ku protein itself forms aggregates, which can bind to the DNA template to form complexes with bright aggregates clearly seen in Fig. 3. Of note, the cryoEM experiments are done with much higher concentrations of protein compared with AFM, so if Ku protein aggregates in solution, one should anticipate large protein aggregates in the cryoEM samples, but these do not appear in Fig. 2. It can be another example of the sample preparation artefact, so the AFM methodology can be used with a high precaution.

3. The height measurements with AFM. The identification of individual Ku proteins in the AFM images is a concern. The images are very complex, so no evidence is provided to convince that these measurements can be assigned to Ku

monomers.

- 4.
5. The lack of evidence for an extensive Ku coating of long DNA templates (Fig. 4). Such coating should be in line with the model proposed. The lack of the long coating suggest that the model is not correct, so additional experiments are required to support the model or modify it. The authors mentioned experiments with long time, so these experiments or/and additional ones with the DNA duplexes with varied lengths should be performed to further validate the model.
6. Cyclization experiments and two types of circles formed. According to the authors' explanation, side-by-side assembly is a transient state for the end-to-end joining type of complexes. Given the high concentrations of DNA and Ku protein, the side-by-side complexes should appear in the CryoEM images. Did the authors see those? If yes, what is the ratio between these two types of complexes? If the end-to-end joining is the final orientation required for the Ku function, the partition between these two states should drop with the increase of the incubation time.

Reviewer #8

(Remarks to the Author)

In this manuscript, Zahid et al. employed cryo-EM, AFM, FIDA, positive-staining imaging to study the oligomerization of Ku-Mtb. The authors resolved structural models of the Ku-Mtb complexes in both the apo and DNA-bound conditions, in which only in the presence of DNA did Ku oligomerize and form filaments. Using mass photometry, FIDA, and AFM, the authors validated the DNA-induced oligomerization of Ku, and quantified the dimensions of the complexes, where DNA-bound Ku oligomers had a mean volume of $\sim 2700\text{nm}^3$ (as judged by both FIDA (Figure 3B) and AFM (Figure 3C)). Disruption of Ku oligomerization by specific mutations was found to impact the mycobacterial survival under DNA damaging conditions. Finally, the authors illustrated the mechanisms of DNA-Ku-Mtb filament formation by further analyzing the cryo-EM structures and performing positive-stain imaging.

I was invited to comment on the AFM part of the study and evaluate the authors' responses regarding questions and concerns raised by other reviewers. In this revision, the authors performed additional AFM experiments, yielding more statistics regarding the filament dimensions, provided more data (with controls), and performed experiments in the cryo-EM equivalent conditions as well as in the absence of divalent ions. These additional experiments and data, in my opinion, resolved the major worry of insufficient statistics as raised by other reviewers. Besides, the authors illustrated their analysis workflow well, with additional comparisons between various conditions, supplementing mounting details on their AFM experiments and data analysis. Therefore, I would like to conclude that this revision have sufficiently addressed the technical issues raised by other reviewers.

Since AFM was performed in aqueous environments, and the substrates had to be absorbed or immobilized onto a surface, the different conformation, as compared to the linear filaments in cryo-EM (substrates floating in the solution) is not unexpected. Indeed, cryo-EM analysis requires averaging of hundreds of thousands of particles while discarding many more particles, which would underestimate the conformational heterogeneity. Thus, the observation of non-linear conformation in AFM, in my opinion, is not a problem, and the dimension measurements are valuable. The authors probably should clearly discuss the limitations of the technique, and elaborate on the reasons for such different observations, especially for people who are not familiar with the technique.

However, I also agree with Review 2 and 7 that AFM in this study provided limited addition to other analyses, under-utilizing the technique. As I mentioned, the dimension measurements are valuable, the oligomerization process induced by DNA is clearly validated, but these observations and conclusion were already drawn from other experiments like FIDA and mass photometry (also in physiologically relevant environments). Therefore, AFM experiments here provide additional evidence, by an alternative technique, rather than offering additional value, in my understanding. Since in this study, AFM does not resolve single molecules, a time-wise experiment, for example, could be performed to provide some kinetic information on the oligomerization process, which would result in great additions to the current work. However, this is not an additional request.

Minor issues:

1. It seems that Fig.3f has never been mentioned in the text
2. In supplementary Fig.11, are these data showing the DNA-bound Ku-Mtb? I think the authors should clearly state that in the legend.

Response to Reviews

Thank you for arranging the review of our manuscript for *Nature Communications*. We appreciate the reviewers' careful evaluation and were pleased to see that they recognized our findings as a significant advancement in the understanding of NHEJ in bacteria.

We have thoroughly addressed the reviewers' comments in our revised submission, and our detailed responses are outlined below. In addition, we have included new data to strengthen the manuscript, including site-directed mutagenesis of the filament interface, supported by both cryo-EM and *in vivo* results, to confirm the importance of key residues. We have also, as requested by several reviewers, repeated and extended the AFM experiments and analysis.

We hope that you will find the revised manuscript substantially improved and now suitable for publication in *Nature Communications*.

We look forward to your response.

Yours sincerely,
Amanda Chaplin and co-authors

REVIEWER COMMENTS

Reviewer #1:

Zahid et al., report the exciting cryo-EM structure of *Mycobacterium tuberculosis* Ku on its own and bound to DNA. Interestingly, they find that Ku forms filaments along the DNA. They propose that in bacterial NHEJ, which lacks DNA-PKcs and other proteins that synapse the DNA double-strand break, Ku filaments are needed to form the stable synapse. The cryo-EM structures also show that the Ku C-terminus is mobile – packed against apo Ku and not visible in the DNA-bound Ku, indicating it has been displaced, potentially in a way to allow for interaction with LigD. Supporting data from AFM, EMSAs, mass photometry, FIDA and positive staining EM confirm Ku forms higher order oligomers when bound to DNA and visualizes previously reported Ku-DNA bridging activity.

This is the first reported structure of the bacterial Ku protein from NHEJ, which will be of interest to those in the DNA repair field, and which provides interesting insights on how bacterial Ku has more functionality than eukaryotic Ku in NHEJ. Previously reported work on bacterial NHEJ has shown biochemically and through single-molecule studies that Ku bridges and synapses the DNA ends. This cryo-EM structure, along with AFM and EM, is the first to visualize it and provide molecular details about the event. This builds on the bacterial NHEJ mechanism and explains how in the absence of proteins found in eukaryotic NHEJ that bridge and synapse the DNA ends, bacterial NHEJ only needs Ku for this DNA-binding activity.

We thank the reviewer for their positive account of the data we present in the manuscript.

Overall, the work is well done. However, the AFM data needs some further detail. Heights and volumes of DNA, Ku and Ku-DNA complexes are reported, but few images are shown and the number of particles measured to get these values are not reported. What was the distribution of particle volumes and heights, to ensure that Ku alone did not form larger oligomers in the absence of DNA? Of minor note – in the Extended Data Figure 8, it says the single Ku (1) and Ku aggregate (2) are 1260 and 4700 nm³, respectively, but in the image and height trace, it appears that the particle labeled as 1 and masked in pink is larger – this may be a mislabel in the figure?

We thank the reviewer for drawing our attention to the AFM data. This Figure has now been replaced in Figure 3. This shows the distribution of the volumes for each Ku-DNA complex (N=349) and DNA-FAM (N=174). The volumes observed for Ku-DNA complexes (median = 273 +/- 729 nm³) are significantly (p-value < 0.0001) large than for Ku alone (median = 69 +/- 358 nm³) or DNA (median = 49 +/- 8.70 nm³), demonstrating clearly that Ku is able to oligomerise in the presence of DNA.

Also, there is only a single image of the Ku-DNA complex shown (Figure 2D). Did all the Ku-DNA complexes appear as similar large aggregates and what kind of distribution was observed in the heights and volumes? Was more than one large aggregate observed and/or were there other smaller Ku-DNA complexes? This would help answer whether the large Ku-DNA aggregate occurs frequently or rarely. Having this data would strengthen the observations and conclusions.

To address this comment and others from across the reviews, we have retaken all the AFM data, working with new protein and DNA stocks to take a larger dataset from which statistical inferences can be made.

We have added Figure 3, we show further examples of large Ku-DNA complexes, with similar extended filamentous conformation. The volume analysis in Figure 3 shows that these complexes are variable in size (IQR = 74 – 895 nm³).

There are a few minor details to address that may also benefit the publication. These are listed below:

1. In the introduction, lines 63-64, it is important to note that while only structures of Mtb ligase and polymerase domains have been solved by x-ray crystallography, there are crystal structures of the LigD phosphoesterase domain as well, but from *P. aeruginosa* (PDB 3N9B). We thank the reviewer for this comment, and we have now included this within the introduction with the corresponding reference.

2. Figure 1A,B – The bars displaying p-value are only between the wildtype Ku strain and the Δ ku pMV306 strain, but not Δ ku. It looks like there should be a significant difference between wildtype and Δ ku as well, but if not, why?

We apologise for the confusion Fig. 1 shows that CFU counts of WT *M. smegmatis* after treatments were significantly higher than CFU counts of delta ku mutant and delta ku mutant containing the empty pMV306 vector (one way ANOVA). We have now added t-test analysis showing that CFU counts of delta ku mutant are significantly lower than CFU counts of WT and replaced this in Fig. 1.

3. In the Extended Data Figure 3, the pLDDT scores, PAE, ipTM and pTM scores should be

reported for the predicted models, especially as the AlphaFold2 model for Ku was used as an initial template for the cryo-EM model building.

We thank the reviewer for this comment and have now included this within a new supplementary Figure 4.

4. It would be interesting to see how well apo and DNA-bound Ku align. Is it only the C-terminus that moves, or do other structural features change as well?

Again, we thank the reviewer for this comment and although we have carried out the overlays, we had not included a figure in the paper – this is now included as a new supplementary Figure 7 And commented on in the text.

5. Lines 291-293 state that the loop interface is more critical than the helix interface in Fig 3D. It is unclear from the written text why this conclusion is made and how the data support this conclusion.

We have now included additional data where we have mutated the helix interface and the loop interface and shown that the loop interface disrupts the filament formation, therefore explaining why this interface is more important. This is shown within Figure 4 C-E as a new panel of data. This shows that L13A/V14A disrupts the filament synaptic Ku interface. The importance of filament formation for DNA repair was further supported by results of *M. smegmatis* *in vivo* experiments (presented in new Fig. 4F, G). The L23A/V24A variant (L23 and V24 residues in *M. smegmatis* Ku corresponding to L13 and V14 in *M. tuberculosis*) did not complement survival defects of the ku deletion mutant after exposure to MMS or desiccation.

Reviewer #2 (Remarks to the Author):

“Oligomerization of Ku from *Mycobacterium Tuberculosis* DNA synapsis.” by S. Zahid et al. presents the biochemical and structural characterization of half the mycobacterium’s two component non-homologous end joining pathway. The key thesis of the proposed research is the pathogenicity of Mtb is related to the genomic integrity maintenance associated with the NHEJ pathway. Therefore, the characterization biochemically and structurally of the Mtb Ku homodimer and its DNA binding will be a significant step in uncovering mechanistic processes that could be targeted for inhibition. The work presented supports this hypothesis by first presenting the role Ku plays in surviving DNA damage and desiccation stress, albeit not essential nor a new finding. Further, this point could benefit from references to supporting research in Mycobacterial NHEJ pathway affecting antibiotic resistance or infectivity.

We thank the reviewer for this comment. Indeed, a double deletion mutant $\Delta ku\Delta ligD$ of *M. tuberculosis* showed a statistically significant defect in infecting of PMA treated monocytic cell line (THP1). We have added this reference to the introduction¹. As far as we are aware NHEJ pathways affect antimicrobial resistance indirectly and further work is required for dissecting this role.

The presentation of the Mtb apo-Ku cryoEM structure demonstrates the similarity between the core protein fold of the heterodimeric Ku70/80 from eukaryotes and the homodimeric bacterial Mtb Ku even though the small size (60 kDa) of the Ku homodimer limited the overall attainable resolution to 4.06 Å. This could likely be improved with lower KeV data collection, (200 KeV versus 300 KeV) to improve the SNR in the particle images leading to

better parameter fitting. However, the high similarity of the Alpha-fold structures to the relatively good resolution cryoTEM structure suggests effort here would be of limited return. We agree that perhaps changing the microscope or collection parameters may have improved the resolution slightly, however given the small size of the homodimer (~60kDa) and as mentioned the similarity to the AlphaFold model, we agree these efforts would likely produce limited improvement with no additional information.

When Ku is bound to a 32 bp dsDNA with a 20 nucleotide poly-T 3' overhang, Ku formed a filamentous structure that enabled cryoEM structure determination to 2.96 Å which was suitable for accurate placement of individual homodimers on dsDNA. Dimer to Dimer interfaces are well resolved in these maps both between strand Ku homodimers and interstrand bound Ku homodimers. AFM experiments were also presented but don't really add any clarity to the assembly as they show poor distribution of the filamentous complex and only show that a consistent height on 8 nm is conserved from Ku alone to Ku-DNA filament.

As mentioned above, we have now repeated all AFM data. We have included new figures in Figure 3 and Supplementary Figures 9, 10 and 11. The data shows that Ku is able to form large extended oligomers in the presence of DNA, which is not observed for Ku alone, or in the absence of DNA (Figure 3 Ci). The median maximum height of the Ku-DNA complexes is 4.37 +/- 3.43 nm (median +/- standard deviation), and they are of length 8 – 25.75 nm (IQR). This can be compared to that of Ku alone which is smaller in height 3.04 +/- 1.63 nm (median +/- standard deviation) and length 5.50 – 9.92 nm (IQR). To compare with our original measurements, we have also traced the Ku-DNA complexes and see a height of around 8 nm as previously observed (Figure 3 Ei).

Further characterization of the Ku DNA filament was carried out by mass photometry and flow induce dispersion analysis which confirmed many of the conclusions of the cryoTEM derived filament structure while providing dissociation and assembly kinetics. A brief description of the filament Ku formation was provided which details the association is driven by hydrophobic helical and loop residues being buried between the dimer interfaces. This would likely generate a highly variable surface that may explain why the filament appears highly variable in the reconstruction process. The structure does not identify any large conformational changes occur from apo-Ku, with the single exception of the c-terminal extended domain which is not resolved but instead lost in the structure when bound to DNA. Presumably, this absence of density is due to displacement similar to the single stranded region of the oligo nucleotide used to build the Ku filament. Finally, larger DNA fragments were bound to Ku and examined by positive staining TEM. Imaging showed recircularization events that varied in end joining topology with the authors stating greater than 80% recircularization by Ku occurred, this data was further quantified in extended data table 2, where a range of 3' ends were examined.

Overall, the paper is concise and coherent adding structure information to the bacterial homologue of eukaryotic Ku70/80. It reconfirms many details of bacterial Ku biochemistry already in the literature for *B. subtilis* and *M. smegmatis* while adding the apo-Ku as well as Ku bound to DNA structures that are at higher resolution, but not significantly more informative <https://doi.org/10.1093/nar/gkac906>. As an NHEJ enthusiast, inclusion of the LigD in the structure determination and biochemical analysis would make for a more comprehensive treatise, minimally deletion of the c-terminal extended domain of Ku to demonstrate their proposed recruitment mechanism for LigD either via rescue of desiccation or MMS sensitivity analysis could easily be performed.

We agree, addition of LigD would indeed yield some interesting results, however we feel this is future/additional work which we are currently undertaking. Over-expression and purification of LigD has proved difficult so would require additional optimisation before structural studies could be attempted. Our pilot data suggest that deletion of the C-terminal extended domain impacts on survival desiccated or MMS-treated *M. smegmatis*, however additional experiments including identification/mutation of critical residues for LigD and Ku interaction are required (and are currently in progress) for definitive conclusions.

The loss of the ssDNA, although easily passed off as being due to the highly flexible nature of ssDNA or maybe a contaminate carried over in the purification thus degrading the ssDNA region raises the question of whether the structure best resolves the blunt end of the oligonucleotide used for assembly or the single strand overhang end. If it is degraded, the resolution of the map is insufficient to determine pyrimidine versus purine on the individual strands to properly fit the sequence, so it is not possible to know. Is the ssDNA region even present in the Ku DNA assembly given the amount of other bands in the purification Mtb Ku (supplementary Fig. 1)?

The cryoEM structure of the blunt ended DNA topology bound to Ku was also solved but if the ssDNA is truly missing this would suggest the structures are essentially the same. Unfortunately, the data in extended data 7 presenting the additional structure appears to not have a PDB or EMDB accession or just wasn't provided to the reviewers.

We apologise and have now adjusted the new supplementary figure 12 to include a PDB fitted to show the similarities between the data collected. As you can see from the data the models are very similar.

To confirm that the single-stranded portion of the DNA utilised is still present AFM studies shown in supplementary figure 9 clearly show that it is still present compared to the blunt DNA. We can therefore be confident of the DNA substrates we are utilising and that filament formation is not due to DNA end configuration. We should also note that in positive EM staining, we did not observe nuclease activity with different types of single-stranded DNA ends, 10 or 40 nm. This therefore also shows that the presence of a contaminant from Ku purification which could digest the ssDNA is unlikely.

DNA end topology (blunt and single stranded were examined but not stem loop or blocked ends) effects on Ku binding and fiber formation would be an interesting addition to this work as the delegation of HR versus NHEJ is growth phase dependent suggesting certain structures may be recognized preferentially. Indeed, the positive staining TEM experiments suggest differences arise in fiber assembly based but bringing this to the cryoEM structural level of insight or more rigorous biochemical level would be of interest.

We thank the reviewer for the suggestion of alternative DNA ends. We have now tried a hairpin DNA substrate and still observe oligomerisation of Ku. We have now included this as additional data in supplementary figure 12. We agree there are further DNA substrates which may be investigated but are not within the scope of this study.

As the manuscript stands, it is abbreviated in its significance to the field but could be improved with exploring further details such as the c-terminal extended domains relevance, possibly exploring the LigD interaction structurally even if only the portion of LigD that interacts with Ku were used, and possibly seeing if focused refinement or variability analysis across the filament helps resolve the more interesting details in the fiber formation. The findings feel superficial and not explored fully given the potential presented within the paper and the previous work in the field.

We thank reviewer for their critical evaluation of our work. In this manuscript we focused on validation of the importance of filament formation for mycobacterial survival (Fig. 4 F, G). As outlined above the role of extended C-terminal domain in mycobacterial survival will be addressed in a separate study.

Minor Points:

1.) Model Angelo results need to be validated as the Ku fiber models have lysine residues that have the atomic coordinates of glycine, those found are position 117 and 184 in KuDNAModel.pdb chains ABEF.

We thank the reviewer for looking at the model so carefully. To confirm these positions are indeed Lys residues but the side chain has just been truncated due to lack of density for the Lys. This is often the case for Lys residues. The PDB deposition sequence alignment was all correct. We can always update and add the side chain back in, if the reviewer feels this is necessary.

2.) Extended figure numbering in the main body text is wrong. Line 186 Extended Figure S7 should refer to S8 and again on line 325 Extended Figure S8 should be S9.

These have now been changed, thank you.

3.) Line 536...FAM-DNA (Reverse) at the 3' end read CCTACA but the complementary strand and structure suggest it should read 3'-CCTTACA-5'

We thank the reviewer for noticing this and have now corrected the sequence.

Reviewer #3 (Remarks to the Author):

Zahid et al. report cryo-EM structures of mycobacterial Ku with and without DNA bound in 4.04 Å and 2.96 Å resolution, respectively. They furthermore show Ku is important for mycobacterial survival under DNA-damaging conditions. The structures reveal that an interesting C-terminal blocking helix prevents filament formation in DNA-free form. Upon DNA binding, the blocking helix is replaced by another Ku heterodimer to form a Ku filament. The DNA end gaps distance spans ~ 40 Å, representing a synaptic complex before LigD repairs breaks. The authors also showed in positive-stained EM that Ku can circularize DNAs and hold DNA ends in proximity. These results are indeed unique in the bacterial NHEJ repair system, and the structures reported show how this works. Readers will also appreciate that authors made several biophysical measurements to understand the biophysical properties of filament formation and stoichiometry.

Ku structural information remains limited, and no structures are available for Ku complexes from any bacterial species. The newly determined DNA-free and bound cryo-EM structures of Ku are an important contribution. The data and insights furthermore have potential for informing bacterial Ku biology and possibly targeting. Before publication the authors should address the following points.

Points.

1. Why does Ku-Mtb need to form filaments along DNA ends? This was not clearly discussed in the text. Human NHEJ proteins can also form filaments (e.g. Hammel et al XRCC4 protein

interactions with XRCC4-like factor (XLF) create an extended grooved scaffold for DNA ligation and double strand break repair. *J Biol Chem.* 2011 Sep 16;286(37):32638-50. doi: 10.1074/jbc.M111.272641; Mahaney et al., XRCC4 and XLF form long helical protein filaments suitable for DNA end protection and alignment to facilitate DNA double strand break repair. *Biochem Cell Biol.* 2013 Feb;91(1):31-41. doi: 10.1139/bcb-2012-0058). Are there relationships or functional analogies here? Readers may wish to relate these results to what is known about the human system.

We thank the reviewer for this comment; we agree more discussion is needed about this and have now included this in the manuscript. We believe filaments of Ku result in protein-protein interactions that help maintain a higher local concentration near the DNA ends. This makes them more stable, both to protect DNA ends and to increase the recruitment of LigD. We have also shown that Ku-*Mtb* synapsis the DNA ends and therefore it is necessary to have a high concentration of protein at the end of the breaks for this important function. In humans Ku70/80 is unable to synapse the DNA without the addition of DNA-PK. We have visualised filaments of DNA-PK in humans², which is intriguing that filaments may play a role in repair in both humans and bacteria but that these assemblies differ. The reviewer is also correct that XRCC4-XLF can form filaments, which is again intriguing, hence why further studies into these differences and nuances must be investigated.

2. Have the authors observed any two Ku-filaments bridging interactions in cryo-EM that mimic the circular end bridging (B1 or B2) in positive stained EM?

This is a good point, however we have not observed any two-Ku filaments bridging in the cryo-EM data such as B1 or B2. We assume to do this, cryo-EM would have to be performed with larger DNA samples, which is not always very favorable for cryo-EM analysis. We have now added some discussion on this.

3. What are the critical structural differences between KuMt and human Ku70/80 that human Ku cannot hold broken DNA ends? The overlay in Fig. S5 is unclear. It should be improved for clarity. One key element is the filament interface – is this similar or not? If not, what is blocking the filament formation in human Ku70/80?

We apologise that this figure is not clear and have now placed the structures side by side rather than overlays to improve clarity, in the new supplementary figure 16. The main difference is Ku-*Mtb* is homodimer compared to human Ku70/80 which is a heterodimer. This difference creates polarity for human Ku70/80 that is not present for Ku-*Mtb*. Apo-Ku-*Mtb* has the C-terminal helix which binds back onto itself and prevents filament formation. A direct overlay of Ku-*Mtb* with human Ku70/80 shows that Ku70/80 already has a helix in this position and therefore the filament interface shown in Ku-*Mtb* is not present in Ku70/80 as shown in supplementary figure 16 and discussed more clearly in the text.

4. While Ku filament formation is a novel observation in bacterial Ku, one of the purposes of understanding Ku-Mtb structures is for therapeutic potential. Can the authors design interface mutants that prevent filament formation and determine whether the interface mutants are critical for mycobacterial survival under damaging agents/stress?

This is a great question, and one which we have been trying to address and can now include in this manuscript. We have found that the key residues are L13 and V14 and that a double mutant of L13A/V14A disrupts the ability of Ku to form oligomers and synapse the DNA ends. We show this through new cryo-EM data, with an example micrograph, 2D classes and a map and model showing two Ku molecules can bind to DNA, but they are no longer able to bridge the DNA break. We have also carried out additional *in vivo* experiments with MMS

and desiccation but with complementation with the L13A/V14A double mutant. Strikingly this double-mutant causes significant survival defects, highlighting the importance of this interface in DNA-repair and bacterial survival during stress conditions. We have now added this new data in Fig. 4.

5. Ku filament formation along DNA ends seems both intriguing and puzzling. For long DNA fragments, how does Ku form filaments along the DNA ends, given that the DNA binding affinity is much tighter than filament formation? Does Ku homodimer slide down the DNA ends, or does the DNA binding channel of Ku open to bind DNA? Are the authors aware of any examples that show Ku can open the DNA binding channel? If not, it seems likely that the sliding event happens. The authors show circularization is independent of DNA end overhang length (Extended Data Table 2), while Robin Oz et al. show 4 nt complementary DNA overhangs increase synapsis lifetime drastically (Nucleic Acids Research 49, 5, 18 March 2021, 2629–2641, <https://doi.org/10.1093/nar/gkab083>). What are authors' views on how Ku forms a filament along DNA ends? This should be concisely noted.

It is known that Ku slides onto the DNA, with this sliding being limited by the interactions between homodimers of Ku forming this filament. In fact, Ku binding to the DNA ends (for Ku-*Mtb* for Ku *B. subtilis* or human Ku70/80), have all been shown to be cooperative. McGovern et al. (2016) doi:[10.1093/nar/gkw149](https://doi.org/10.1093/nar/gkw149) also showed, by positive EM staining, that Ku *subtilis* formed oligomers at DNA ends. Ku interacts with the DNA end and then translocate inside the DNA. It is the protein-protein interactions we define in detail within this manuscript that allow them to be held close to the DNA end. This is indeed a sliding that is facilitated depending on the size of the central ring. Although we show that circularization is independent of DNA end overhang length, the synaptic lifetime may differ as shown in NAR 2021 mentioned. Since bridging events are dynamic, it is normal that the presence of complementary sequences increases lifespan. We have now added further discussion of filament formation.

6. Do the authors see any structure differences between the DNA gap interface and the DNA engaged interface?

As mentioned in the text the only structural difference we see is that the helices containing residues Val198, Met202 and Met194 are displaced relative to each other at the junction between minimal repeating units (differing whether the interface is a gap in the DNA or engaging DNA). Whereas the loop containing residues Leu13 and Val14 are consistent throughout the filament. It is these loop residues in which we have mutated and shown to be critical for filament formation and synapsis.

7. Line 185 should be Extended Data Fig S8. Not S7.

This has now been changed, thank you.

8. For readers to follow a line of logic, the figures should be cited in order instead of randomly.

The figures have been modified to include additional data and therefore the logical order has been revised and checked.

Reviewer #4 (Remarks to the Author): (Identical to reviewer 3?).

Zahid et al. reported cryo-EM structures of mycobacterial Ku with and without DNA bound

in 4.04 Å and 2.96 Å resolution, respectively. They also showed Ku is important for mycobacterial survival under DNA-damaging conditions. The structures revealed a C-terminal blocking helix prevents filament formation in DNA-free form. Upon DNA binding, a blocking helix is replaced by another Ku heterodimer to form a Ku filament. The DNA end gaps distance ~ 40 Å, representing a synaptic complex before LigD repairing the breaks. They also showed in positive-stained EM that Ku can circularize DNAs and hold DNA ends in proximity. This is indeed unique in the bacterial NHEJ repair system, and the structures here show how. I also appreciate that authors made several biophysical measurements to understand the biophysical properties of filament formation and stoichiometry.

Specific comments.

1. Why does Ku-Mtb need to form filaments along DNA ends? It was not clearly discussed in the text.
2. Have authors observed two Ku-filaments bridging interaction in cryoEM mimicking the circular end bridging (B1 or B2) in positive stained EM?
3. What are the critical structural differences between KuMt and human Ku70/80 that human Ku cannot hold broken DNA ends? The overlay in Fig. S5 is unclear and needs improvement for clarity. One key element is the filament interface, similar or not? If not, what is blocking the filament formation in human Ku70/80?
4. While Ku filament formation is a novel observation in bacterial Ku, one of the purposes of understanding Ku-Mtb structures is for therapeutic potential. Can authors design interface mutants that prevent filament formation and determine whether the interface mutants are critical for mycobacterial survival under damaging agents/stress?
5. It is intriguing and puzzling about Ku filament formation along DNA ends. For long DNA fragments, how does Ku form filaments along the DNA ends, given that the DNA binding affinity is much tighter than filament formation? Does Ku homodimer slide down the DNA ends, or does the DNA binding channel of Ku open to bind DNA? I haven't seen any examples that show Ku has the ability to open the DNA binding channel. So it is likely the slide event happens. The authors show circularization is independent of DNA end overhang length (Extended Data Table 2), while Robin Oz et al. show 4 nt complementary DNA overhangs increase synapsis lifetime drastically (Nucleic Acids Research, Volume 49, Issue 5, 18 March 2021, Pages 2629–2641, <https://doi.org/10.1093/nar/gkab083>). What are authors' opinions on how Ku forms a filament along DNA ends?
6. Do authors see the structure difference between the DNA gap interface and the DNA engaged interface?
7. Line 185 should be Extended Data Fig S8. Not S7. And it would be great if the figures could be followed in order instead of randomly.

Reviewer #5 (Remarks to the Author):

The authors present the results of extensive structural studies of complexes with DNA Ku protein of *Mycobacterium tuberculosis* bacteria. Similar to Ku protein of vertebrates, the bacterial Ku protein is critically needed for the non-homologous end-joining (NHEJ) DNA repair, however process can be different as multiple proteins are involved in the NHEJ process for vertebrates. The authors applied multiple experimental approaches to characterize the complexes of Ku protein with DNA substrates and found the oligomerization of the

protein bound to DNA is the key factor requiring for the synapsis of the DNA ends. CryoEM data led the authors to the model of the end-joining with the key role of the C-terminus of Ku in this process. However, a number of issues need to be clarified to make a convincing case.

1. The authors used a hybrid DNA substrate with 32 bp DNA duplex tailed with 20 nt ssDNA in the majority of their studies including high-resolution cryo-EM studies. However, there is no indication in the CryoEM data to the ability of Ku protein binding ssDNA. Only two Ku dimers per 32 bp DNA duplex. What is the role of ssDNA tail in the complex assembly? It not identified in the cryo-EM data.

This specific DNA was initially used due to the paper on UvrD1 indicating an interaction between UvrD1 and Ku with this specific DNA³. We have however also utilised blunt DNA and hairpin DNA as shown in supplementary figure 12 and revealed that DNA end configuration is not important for synapsis and oligomerisation. Therefore, although we used DNA with a ss overhang as shown in our cryo-EM data, AFM and positive stain EM this configuration is not important. We have now made that clear in the text.

2. AFM data are not in line with EM data including CryoEM ones. First, picture in Fig 2D shows a particle with curved Y-shape, whereas the Cryo EM data in Fig 2C show linear filaments. If the images of complexes are different and in-line with cryo-EM data, these should be presented, and the population of various assemblies should be characterized. We have repeated the AFM data as mentioned above and in response to this and other reviewers and now included new figures, 3 and supplementary figures 9-11. The data shows that Ku is able to form large extended oligomers in the presence of DNA, which are not observed for Ku alone or in the absence of DNA.

3. AFM volume measurements – they suggest that almost 100 Ku protein are bound. Apparently more than one DNA template was in that assembly. Are these typical data? Was the sample the same as used for CryoEM studies? Again only linear filaments are shown in Fig. 2C – did such filaments seen in the AFM data?

As mentioned, we have repeated the AFM data. We observe a range of large filamentous constructs for Ku co-incubated with DNA (Figure 3). The samples were prepared in the same ways as for cryo-EM experiments with a 1:4 molar ratio of DNA:Ku co-incubated in the same buffer (75 mM NaCl, 5 mM MgCl₂, 20 mM Tris, pH 7.4). We have included additional statistical analysis and further examples of this data is shown in Figure 3 and supplementary figures 9-11. From these data we agree that ~100 Ku proteins could be contained within these complexes but also observe that these will contain DNA constructs in addition to the protein.

4. The height measurements in Fig 2D – only one cross-section is shown – what are other measurements? The analysis over the statistically significant set of images to support the conclusion is missing.

We have now shown a range of AFM images of Ku-DNA complexes, which take large extended filamentous forms (Figure 3). We have characterised these by their height, length and volume (supplementary figures 9-11), showing for each that there is a significant difference between Ku alone and Ku-DNA complexes.

5. The model of the Ku-DNA complexes suggests a coating of the DNA duplex with the protein with a high density – two dimers per three DNA turns. Why only a limited DNA segment is coated with proteins with experiments with long DNA substrates (Fig. 4) with a high concentration of the protein?

We do indeed observe limited coverage of DNA ends by Ku. Perhaps increasing the incubation time as well as the Ku concentration may increase this coated segment at DNA ends.

6. DNA cyclization by Ku protein. According to Fig. 4, two types of circles are formed. Apparently only one of them is an appropriate substrate for LigD ligase. What is the ratio between the two types of DNA rings depending on the Ku protein concentration? Could it be that the right type of rings is formed at lower concentration of the protein? What would be the Ku stoichiometry for the rings formed at lower concentrations of Ku? Experiments with lower Ku concentration are needed to clarify this physiologically important property of Ku. It's likely that bridging is more conducive to Ligase D's action because the end-to-end configuration makes it difficult for Ligase D to access the ends. In fact, we must consider these two types of configurations: end-to-end and bridging (b1 or b2) are equilibrium states that allow transition from one to the other. Moreover, when we decrease the concentration, we simply decrease the number of complexes relative to the free ones. Ku binding is cooperative, and we have an average of 3 to 4 Ku complexes per end. Thus, when we go down to 30 nM Ku, we don't have enough complexes to be significant. We tested 30 nM/50 nM, 100 nM, and 200 nM. It's likely that bridging is the first step before end-to-end. There's an equilibrium between them

Reviewer #6 (Remarks to the Author):

Reviewer #7 (Remarks to the Author):

This manuscript presents a structural and functional analysis of Ku from Mycobacterium tuberculosis (Ku-Mtb). They demonstrate its ability to form oligomers on DNA and support DNA end synapsis during non-homologous end joining (NHEJ). The authors combine cryo-EM, mass photometry, AFM, positive stain EM, and biochemical assays to reveal that Ku-Mtb filaments coat DNA and facilitate recruitment of LigD for ligation, bypassing the need for DNA-PKcs present in eukaryotic NHEJ systems. Importantly, they identify the displacement of a C-terminal α -helix during filament formation and propose a model where this movement promotes LigD recruitment.

This work is original, comprehensive, and timely, as it sheds light on bacterial NHEJ mechanisms. The structural insights provided are novel, particularly in highlighting filament formation by Ku-Mtb—a behavior not observed in human Ku70/80. The study is of broad interest to structural biologists, microbiologists, and researchers in DNA repair. Overall, the EM, EMSA and mass photometry data appear to be of high quality; however the AFM data shown in the paper is minimal, although there is discussion of population analysis, no data are shown, and perhaps more concerning the AFM experiments were conducted at extremely low salt, bringing into question the physiological relevance of these experiments.

We agree with the reviewer entirely and have performed new AFM imaging experiments, incubating and immobilising our complexes in the same buffer that was used for the cryo-EM

experiments, these were then rinsed in NiCl_2 for in liquid imaging. To validate the experiments, an additional set of experiments were performed using APS immobilisation in the absence of divalent ions (75 mM NaCl, 20 mM Tris, pH 7.4). We have included these data in the manuscript and supplementary information and rewritten the methods section of the manuscript to reflect both the new experiments and also the additional statistical analysis performed.

Major Issues:

1. AFM Experimental Limitations (Data Presentation, Replication, Controls)

o The AFM analysis is a critical part of the study but lacks essential detail and rigor. The authors show height profiles from only a single trace per condition without reporting the number of molecules analyzed (N), number of replicates, or any statistical variation (e.g., standard deviation or SEM). No histograms or frequency distributions of volume measurements are provided, making it difficult to assess reproducibility or heterogeneity. The authors should present distributions and include the number of complexes analyzed. The new AFM data (Figures 3 and supplementary figures 9-11) contains data from 3 experimental replicates, with the data for each experiment combined into a single combined strip and violin plot. Each figure explicitly states the number of molecules, mean and standard error for each sample type. In addition, we have performed statistical testing to determine significance between each sample type, showing a significant difference between Ku and Ku-DNA for all measurements and sample conditions.

o The authors show only a single complex which appears relatively large. This large complex would not be expected based on their EMSA studies. It is likely that the large complexes are an artifact of the very low salt and high divalent ion concentration (25mM MgCl_2 , 10mM Tris pH 7.4). It is likely that the authors are using these conditions to keep the complexes attached for solution imaging; however there are other choices (e.g., poly-L-lysine, APTES, or even native Mg^{2+} adsorption) that allow physiological salt concentrations to be used. It is unclear why the authors chose solution imaging, which is much more difficult than air imaging, when the data the authors desire could be easily obtained using air imaging. The authors underutilize the potential of in-liquid AFM. For example, they could support their filament model with dynamic imaging (movies), conformational change analysis, or bend angle quantification, which are all feasible and would strengthen their claims about higher-order oligomeric states and synapsis.

We chose solution imaging as it is the best to observe the small DNA constructs, including ssDNA overhang and the Ku proteins in a hydrated state. We do not perform dynamic imaging but have performed multiple experimental replicates and statistical analysis of the complexes contained. We have compared the conformational changes between Ku protein and DNA alone and complexed together seeing significant differences between them. Finally, we have performed tracing of large complexes which tracks the filamentous backbone and shows comparable data to our previous results.

o There is no mention of appropriate AFM controls. Without showing DNA-only and Ku-only traces in parallel under matched imaging conditions and buffer, it is hard to distinguish specific Ku-DNA filament formation from nonspecific aggregation.

These have now been included as specified above.

2. Positive stain EM classification criteria

o Similar to the AFM studies, the authors do not show distributions of species and do not give

the number of complexes analyzed. This analysis would benefit from a length distribution graph for the linear and circular structures

This information has now been included within supplementary Table 2.

3. C-terminal α -helix movement and LigD recruitment (interpretation clarity)

o While the proposed model for C-terminal α -helix displacement and LigD recruitment is intriguing, it is speculative. No direct evidence for LigD interaction with Ku-Mtb in this study is shown. The authors should clarify what is directly supported by cryo-EM versus inferred from prior work. Indicate that LigD binding remains to be validated experimentally. We thank the reviewer for this comment and have now made it clear in the text and within Figure 6.

Minor Issues:

1. The numbering of extended figures is inconsistent throughout the text (e.g., Extended Data Fig. 7 is actually Fig. 8). This has now been changed, thank you.

2. Figure 5 (model) could be improved by clearly distinguishing experimentally validated structures from modeled components (e.g., LigD).

We have now changed this figure to label it more clearly that LigD is just a model and not experimentally determined.

3. Textual clarity would benefit from paragraph spacing in the Discussion to ease readability. We have now revised this.

4. Clarify if Ku-Mtb expression levels (e.g., stationary phase) were quantified or inferred. Levels of ku expression were not tested in this study. Previously published data⁴ showed that levels of Ku protein did not change significantly in *M. smegmatis* at different growth stages, including stationary phase.

5. Minor typos (e.g., "Cterminal" instead of "C-terminal" in figure legends) should be corrected.

This has been checked and corrected.

References:

- 1 Brzostek, A., Szulc, I., Klink, M., Brzezinska, M., Sulowska, Z. & Dziadek, J. *PLoS One* **9**, e92799, (2014).
- 2 Hardwick, S. W., Stavridi, A. K., Chirgadze, D. Y., De Oliveira, T. M., Charbonnier, J. B., Ropars, V., Meek, K., Blundell, T. L. & Chaplin, A. K. *Structure* **31**, 895-902 e893, (2023).
- 3 Chadda, A., Jensen, D., Tomko, E. J., Ruiz Manzano, A., Nguyen, B., Lohman, T. M. & Galburt, E. A. *Proc Natl Acad Sci U S A* **119**, (2022).
- 4 Zhou, Y., Chen, T., Zhou, L., Fleming, J., Deng, J., Wang, X., Wang, L., Wang, Y., Zhang, X., Wei, W. *et al. FEMS Microbiol Lett* **362**, (2015).

Response to Reviewers 2

We thank the editor and reviewers for their thorough and thoughtful evaluation of our manuscript. We are encouraged by the overall positive feedback and pleased that the reviewers consider our work a significant contribution to the field and suitable for publication. Below, we provide a detailed point-by-point response to the remaining comments.

REVIEWERS' COMMENTS

Reviewer #1 (Remarks to the Author):

The authors have addressed all previous reviewer comments from the initial review successfully. Two minor points to consider are included below when making final edits for publication, otherwise, the article is highly significant for the field, with the work supporting the conclusions drawn and has used appropriate methodology to achieve these results.

Minor points:

1. For supplementary figure 7, please include an RMSD for the aligned structures to provide a quantitative measure.

This has now been checked and included.

2. On page 10, line 261, nm should be nm³.

This has now been changed.

Reviewer #2 (Remarks to the Author):

“Oligomerization of Ku from Mycobacterium tuberculosis DNA synapsis.” by S. Zahid et al. has improved the manuscript significantly.

Of interest is the improved statistics presented for the AFM and positive stain DNA topology experiments.

In support of their hypothesis for the c-terminal helices role in filament or synapse formation, mutations to two residues L13A and V14A proximal the helices but within the dimer interface were generated that disrupted filament formation as assessed cryoTEM structure analysis. The c-terminal protomer helices still look to be displaced in the DNA bound KuMTB mutant cryoTEM structure, but is not discussed. On line 312 of the revised manuscript the authors' state, “Mutations introduced at the helical interface did not disrupt oligomer formation.” The nature of these mutation attempts were not detailed in the manuscript and are still useful information regardless of the negative outcomes, additionally how it was determined these mutations failed to disrupt filament formation would also be useful information.

We thank the reviewer for this comment and agree it is important information about how we decided these mutations did not disrupt filament formation. We have now included text to describe how we purified the double mutation at the helix interface and carried out cryo-EM analysis. We have also included an additional new **Supplementary Figure 12.**

Additionally, the authors directly address DNA end configuration and filament formation and KuMTB binding. Together with the mutational analysis, this is a

significant addition to the understanding of the filament formation process by the MTB Ku DNA complex.

Due to these revision, the work is sufficiently improved and has significant relevance to multiple fields to warrant consideration for publication.

We thank the reviewer for the positive comments.

Reviewer #3 (Remarks to the Author):

The revised manuscript is suitable for publication. However, one added modification is needed to the sentence “Invertebrates including, bacteria and yeast do not contain DNA-PKcs-like enzymes meaning an enormous number of organisms (more than the number of vertebrate organisms) carry out NHEJ without DNA-PKcs.” This statement should be updated in the light of the 2021 publication noting “...DNA-PKcs is widely distributed in invertebrates, fungi, plants, and protists...” and this publication should be cited (Lees-Miller JP, et al.,. Uncovering DNA-PKcs ancient phylogeny, unique sequence motifs and insights for human disease. *Prog Biophys Mol Biol.* 2021 Aug;163:87-108. doi: 10.1016/j.pbiomolbio.2020.09.010).

This revised manuscript is a significant contribution to the field, and will be of general interest. The authors created new interface mutants (L13A/V14A) and have found the double mutant disrupted the Ku-Mtb filament formation and ability to bind synaptic DNA ends, as well as decreased cell survival under DNA-damaging conditions. They also added new figures to improve the clarity to support their model and conclusion.

The implications of these findings could extend to understanding how similar mechanisms operate in other organisms, such as humans. Additionally, further exploration of the mutant's effects could pave the way for novel therapeutic strategies targeting DNA repair pathways in *Mycobacterium tuberculosis*.

Reviewer #4 (Remarks to the Author):

The authors have addressed all my concerns in their rebuttal. They created new interface mutants (L13A/V14A) and have found the double mutant disrupted the Ku-Mtb filament formation and ability to bind synaptic DNA ends, as well as decreased cell survival under DNA-damaging conditions. They also added new figures to improve the clarity to support their model and conclusion. This is a significant contribution to the field, and it would be great to see this paper published. The implications of these findings could extend to understanding how similar mechanisms operate in other organisms, such as humans. Additionally, further exploration of the mutant's effects could pave the way for novel therapeutic strategies targeting DNA repair pathways in *Mycobacterium tuberculosis*.

We thank the reviewer for the positive comments.

Reviewer #5 (Remarks to the Author):

The authors' responses are not satisfactory. I keep my comments in the same order as originally.

We apologise if we have not responded sufficiently to the reviewers concerns and aim to address these further below.

1. The use of the hybrid DNA template with 32 bp DNA duplex tailed with 20 nt ssDNA. The KU-mediated template assembly with such an asymmetric monomer will lead to three types of the dimers: blunt end-DNA duplex- ssDNA+ssDNA-DNA duplex-blunt end; blunt end-DNA duplex- ssDNA+blunt end-DNA duplex- ssDNA; ssDNA-DNA duplex-blunt end+ blunt end-DNA duplex- ssDNA. However, only the dimer with the structure (blunt end-DNA duplex-blunt end+ blunt end-DNA duplex-blunt end) is presented as a model based on structural studies with cryoEM. However, no such a template with blunt ends is used. Although a visualization of ssDNA tail with cryoEM can be a problem, the tail is large and can contribute to the monomer's assembly in the Ku-DNA arrays. Moreover, the 4 nm distance between the DNA ends in the model is close to the size of ssDNA coil with 20 residues. If there is no Ku binding to these oligoT ends, the ssDNA segments can work as spacers between the DNA templates. Additionally, assembly of dimers and higher oligomers via ssDNA tails can lead to branched structures observed in the cryoEM images. Unfortunately, this fundamental problem is not discussed in the paper nor addressed by the authors to my original concern. Given that the major function of Ku protein is to join the DNA ends, primarily with the blunt end morphology, the experiments with DNA template with blunt ends are required to validate/test the model proposed by the authors.

We agree with the reviewers concerns about the single-stranded segment of the DNA used in the majority of the experiments. However, as shown in the manuscript in new **Supplementary Figure 13**, we have repeated the cryo-EM experiments using blunt DNA and produced identical results. These experiments although only presented in the supplementary are very time consuming and expensive and should not be over-looked. The fact that we reveal identical oligomerisation of Ku on blunt DNA does indeed tell us that the ssDNA portion of the DNA is not important in oligomerisation, nor is it important in the distance between the Ku molecules as suggested. We have now looked back at our data to double-check we do not see any branched structures and can confirm that we do not. We have now discussed this in the text. We are unsure what further experiments the reviewer would like to see.

2. CryoEM vs AFM data. These are fundamentally different images. Large complexes with no clear linear arrays appear in all AFM images in the manuscript. The difference between the CryoEM vs AFM results is apparently due to the differences in the sample preparation procedures. In AFM, the samples are prepared at the liquid-surface interface, which can contribute to the higher order assembly of the DNA-protein complexes. The use of the DNA template with ssDNA tail can be an additional factor contributing to clustering observed with AFM. Importantly, even Ku protein itself forms aggregates, which can bind to the DNA template to form complexes with bright aggregates clearly seen in Fig. 3. Of note, the cryoEM experiments are done with much higher concentrations of protein compared with AFM, so if Ku protein aggregates in solution, one should anticipate large protein aggregates in the cryoEM samples, but these do not appear in Fig. 2. It can be

another example of the sample preparation artefact, so the AFM methodology can be used with a high precaution.

We have now added additional text on the differences between AFM and cryo-EM.

3. The height measurements with AFM. The identification of individual Ku proteins in the AFM images is a concern. The images are very complex, so no evidence is provided to convince that these measurements can be assigned to Ku monomers. We have now explored this data with caution.

4.

5. The lack of evidence for an extensive Ku coating of long DNA templates (Fig. 4). Such coating should be in line with the model proposed. The lack of the long coating suggest that the model is not correct, so additional experiments are required to support the model or modify it. The authors mentioned experiments with long time, so these experiments or/and additional ones with the DNA duplexes with varied lengths should be performed to further validate the model.

Just to check with reviewer 5 that Fig.4 in the update manuscript is about the filament interface. Just to check the reviewer means Fig. 5 – positive stain. We believe our experiments show the ability of Ku to coat the DNA ends and that is the importance of this study, further exploration of the nature of this oligomerisation will be explored in future studies.

6. Cyclization experiments and two types of circles formed. According to the authors' explanation, side-by-side assembly is a transient state for the end-to-end joining type of complexes. Given the high concentrations of DNA and Ku protein, the side-by-side complexes should appear in the CryoEM images. Did the authors see those? If yes, what is the ratio between these two types of complexes? If the end-to-end joining is the final orientation required for the Ku function, the partition between these two states should drop with the increase of the incubation time.

We do not see these types of assemblies as mentioned previously and included in the manuscript text. Future experiments to alter concentrations etc may allow for visualisation.

Reviewer #8 (Remarks to the Author):

In this manuscript, Zahid et al. employed cryo-EM, AFM, FIDA, positive-staining imaging to study the oligomerization of Ku-Mtb. The authors resolved structural models of the Ku-Mtb complexes in both the apo and DNA-bound conditions, in which only in the presence of DNA did Ku oligomerize and form filaments. Using mass photometry, FIDA, and AFM, the authors validated the DNA-induced oligomerization of Ku, and quantified the dimensions of the complexes, where DNA-bound Ku oligomers had a mean volume of $\sim 2700\text{nm}^3$ (as judged by both FIDA (Figure 3B) and AFM (Figure 3C)). Disruption of Ku oligomerization by specific mutations was found to impact the mycobacterial survival under DNA damaging conditions. Finally, the authors illustrated the mechanisms of DNA-Ku-Mtb filament formation by further analyzing the cryo-EM structures and performing positive-stain imaging.

I was invited to comment on the AFM part of the study and evaluate the authors'

responses regarding questions and concerns raised by other reviewers. In this revision, the authors performed additional AFM experiments, yielding more statistics regarding the filament dimensions, provided more data (with controls), and performed experiments in the cryo-EM equivalent conditions as well as in the absence of divalent ions. These additional experiments and data, in my opinion, resolved the major worry of insufficient statistics as raised by other reviewers.

We thank the reviewer's response that the additional experiments addressed previous concerns.

Besides, the authors illustrated their analysis workflow well, with additional comparisons between various conditions, supplementing mounting details on their AFM experiments and data analysis. Therefore, I would like to conclude that this revision have sufficiently addressed the technical issues raised by other reviewers. Since AFM was performed in aqueous environments, and the substrates had to be absorbed or immobilized onto a surface, the different conformation, as compared to the linear filaments in cryo-EM (substrates floating in the solution) is not unexpected. Indeed, cryo-EM analysis requires averaging of hundreds of thousands of particles while discarding many more particles, which would underestimate the conformational heterogeneity. Thus, the observation of non-linear conformation in AFM, in my opinion, is not a problem, and the dimension measurements are valuable. The authors probably should clearly discuss the limitations of the technique, and elaborate on the reasons for such different observations, especially for people who are not familiar with the technique.

We have now added additional text to discuss this.

However, I also agree with Review 2 and 7 that AFM in this study provided limited addition to other analyses, under-utilizing the technique. As I mentioned, the dimension measurements are valuable, the oligomerization process induced by DNA is clearly validated, but these observations and conclusion were already drawn from other experiments like FIDA and mass photometry (also in physiologically relevant environments). Therefore, AFM experiments here provide additional evidence, by an alternative technique, rather than offering additional value, in my understanding. Since in this study, AFM does not resolve single molecules, a time-wise experiment, for example, could be performed to provide some kinetic information on the oligomerization process, which would result in great additions to the current work. However, this is not an additional request.

We agree the AFM experiments are complementary to the other techniques, and we have added text to discuss this.

Minor issues:

1. It seems that Fig.3f has never been mentioned in the text

Thank you- this has now been added.

2. In supplementary Fig.11, are these data showing the DNA-bound Ku-Mtb? I think the authors should clearly state that in the legend.

This has now been clearly stated.